# Secreted protease PRSS35 suppresses hepatocellular carcinoma by disabling CXCL2-mediated neutrophil extracellular traps

Ting Wang[1,2,6], Yingli Zhou[1,2,6], Zilong Zhou[1,2,6], Pinggen Zhang[1,2], Ronghui Yan[1,2], Linchong Sun [3], Wenhao Ma[1,2], Tong Zhang[3], Shengqi Shen[3], Haiying Liu[1,2], Hui Lu[1,2], Ling Ye[1,2], Junru Feng[1,2], Zhaolin Chen[1], Xiuying Zhong[3], Gao Wu[2], Yongping Cai[4], Weidong Jia[1], Ping Gao [2,3] ✉ & Huafeng Zhang [1,2,5] ✉

Hepatocytes function largely through the secretion of proteins that regulate cell proliferation, metabolism, and intercellular communications. During the progression of hepatocellular carcinoma (HCC), the hepatocyte secretome changes dynamically as both a consequence and a causative factor in tumorigenesis, although the full scope of secreted protein function in this process remains unclear. Here, we show that the secreted pseudo serine protease PRSS35 functions as a tumor suppressor in HCC. Mechanistically, we demonstrate that active PRSS35 is processed via cleavage by proprotein convertases. Active PRSS35 then suppresses protein levels of CXCL2 through targeted cleavage of tandem lysine (KK) recognition motif. Consequently, CXCL2 degradation attenuates neutrophil recruitment to tumors and formation of neutrophil extracellular traps, ultimately suppressing HCC progression. These findings expand our understanding of the hepatocyte secretome's role in cancer development while providing a basis for the clinical translation of PRRS35 as a therapeutic target or diagnostic biomarker.

During the development of hepatocellular carcinoma (HCC), the expression of numerous proteins is altered in hepatocytes or HCC cells to facilitate tumor cell survival and proliferation[1,2]. As a major component of the liver proteome, the secretome of hepatocytes or HCC cells can also contribute determining factors in cancer progression[3,4]. In recent years, several secreted proteins have emerged as diagnostic markers or treatment targets for various types of cancers, which are still undergoing validation in clinical trials[5]. However, considerable gaps persist in our understanding of the scope and mechanisms of secreted proteins that participate in HCC pathogenesis, which could

greatly advance the development of markers for early diagnosis and effective therapeutic strategies for this deadly disease.

Several proteins have been established as tumor suppressors, and the loss of their function can directly regulate the proliferation and functions of different cancer cells[6,7]. With the development of cancer immunity, tumor suppressors also regulate leukocytes, including immune cells in the tumor microenvironment (TME), to mediate strong, indirect suppression of tumor progression[8,9]. As a major subtype of leukocyte, neutrophils have been discovered as prevalent members of the TME, where they exert dual functions. On one hand,

[1]Anhui Key Laboratory of Hepatopancreatobiliary Surgery, Department of General Surgery, Anhui Provincial Hospital, the First Affiliated Hospital of USTC, Division of Life Science and Medicine, University of Science and Technology of China, Hefei, China. [2]The Chinese Academy of Sciences Key Laboratory of Innate Immunity and Chronic Disease, School of Basic Medical Sciences, Division of Life Science and Medicine, University of Science and Technology of China, Hefei, China. [3]Medical Research Institute, Guangdong Provincial People's Hospital, Guangdong Academy of Medical Sciences, Southern Medical University, Guangzhou, China. [4]Department of Pathology, School of Medicine, Anhui Medical University, Hefei, China. [5]Anhui Province Key Laboratory of Biomedical Aging Research, Division of Life Science and Medicine, University of Science and Technology of China, Hefei, China. [6]These authors contributed equally: Ting Wang, Yingli Zhou, Zilong Zhou. ✉e-mail: pgao2@ustc.edu.cn; hzhang22@ustc.edu.cn

neutrophils mediate antitumor responses through the direct killing of tumor cells and by participating in cellular networks that mediate antitumor resistance. On the other hand, neutrophils also activate an inflammatory response that can promote tumor growth by driving angiogenesis, extracellular matrix remodeling, metastasis, and immunosuppression. Whether neutrophils function in a tumor-suppressive or tumor-promoting capacity depends on factors in the TME as well as their own diversity and plasticity[10–12]. While neutrophils affect tumor progression through multiple mechanisms, evidence is also currently emerging for a role of neutrophil extracellular traps (NETs), a product released by neutrophils[13]. NETs are large extracellular web-like structures consisting of chromatin DNA filaments coated with granule proteins, that may prevent or limit infection[14,15] by trapping and immobilizing bacteria[16], fungi[17], viruses[18] and parasites[19] so that they can be eliminated by other secreted anti-microbial compounds. The dysregulation of NETs formation or activity has been shown to drive the development of cancer and other immune-related diseases[13]. Nevertheless, our understanding of NETs involvement in cancer development is still in the infant stage.

As a major proportion of the non-cellular components in the tumor microenvironment, secreted proteins serve as a means of intercellular communication between host leukocytes and tumor cells, some of which can regulate neutrophil behavior in the TME. Here, by secretome analysis of hepatocyte and liver cancer cell lines, we identify a secreted tumor suppressor, PRSS35, that inhibits HCC progression through cleavage of the chemokine CXCL2, which mediates pro-tumor neutrophil function. The findings described here expand the scope of our understanding of the role of interplay between secreted proteins and neutrophils in HCC development, and suggest a potential diagnostic marker and therapeutic target for this disease.

## Results

### PRSS35 protein abundance is decreased in the HCC secretome

To identify potential factors related to the development and progression of HCC, we employed label-free proteomic analysis of the HCC secretome and compared the protein profiles of human PLC liver cancer cells with that of human THLE3 hepatocytes secreted in conditioned medium. We found that many proteins showed differential abundance in the conditioned medium used for PLC cell culture compared to that of THLE3 cells (Supplementary Fig. 1a, left panel), among which 236 proteins were designated as secreted proteins in the UniProt database[20]. Analysis of these 236 secreted proteins revealed that PRSS35 was the most significantly downregulated protein in the PLC secretome (Fig. 1a, Supplementary Fig. 1a, right panel and Supplementary Data 2). Western blot (WB) analysis using an antibody that recognizes the N-terminus of PRSS35 indicated that both intracellular and extracellular PRSS35 protein levels were markedly reduced in PLC, HepG2, and Hep3B liver cancer cells, relative to its accumulation in THLE3 cells (Fig. 1b). Interestingly, the molecular weight of PRSS35 enriched in culture medium (SF-PRSS35) was far lower than that of full-length PRSS35 (FL-PRSS35) isolated from cell lysates (Fig. 1b). Overexpression of PRSS35 in PLC, HepG2, and Hep3B liver cancer cells further led to the identification of a short form (SF-PRSS35) in the culture medium (Supplementary Fig. 1b), suggesting that this short form was specifically secreted. Further WB analysis using antibodies targeted to different peptide regions of PRSS35 (i.e., designated N-PRSS35, M-PRSS35, and C-PRSS35 based on their respective antigen sequences, Supplementary Fig. 1c) revealed that multiple short forms of PRSS35 protein were enriched in the culture medium of PLC cells overexpressing PRSS35 (Fig. 1c). Mass spectrometry (MS) also confirmed that the shorter bands detected by SDS-PAGE indeed included several PRSS35 peptides (Fig. 1d and Supplementary Data 3). Collectively, these results identified PRSS35 as a secreted protein with significantly lower abundance in HCC cells, potentially due to cleavage into multiple fragments.

We then used WB to detect full-length PRSS35 and its shorter variants in clinical samples of HCC patients, which revealed that all forms of this protein were markedly decreased in HCC lesions compared to their levels in adjacent noncancerous tissues (Supplementary Fig. 1d). Subsequent immunohistochemistry (IHC) analysis using C-PRSS35 antibody also showed a gradual reduction in PRSS35 levels with increasing stage of HCC development (Supplementary Fig. 1e and Supplementary Tables 1, 2). In addition, patients expressing high PRSS35 in their HCC lesions exhibited much longer survival times than those with low PRSS35 expression (Supplementary Fig. 1f and Supplementary Table 3). More importantly, the levels of different truncated PRSS35 forms, but not full-length PRSS35, were markedly decreased in HCC patient serum compared to that in normal subjects (Fig. 1e). To further quantify PRSS35 levels in patient serum, we developed a customized PRSS35 ELISA kit using two antibodies against the N terminal region (Supplementary Fig. 1g, h). ELISA analysis of serum from 149 HCC patients and 73 normal subjects revealed significantly lower serum levels of PRSS35 in HCC patients than that in normal subjects (Fig. 1f). These results suggested that secreted PRSS35 protein could serve as a potential prognostic biomarker for HCC patients. Furthermore, we predicted the possible transcriptional factor(s) regulating PRSS35 by JASPAR database, and found that HNF4A is the most potential candidate. Furthermore, we observed that both PRSS35 protein and mRNA levels were downregulated when we knocked down HNF4A in HepG2 cells, and up-regulated with HNF4A overexpression (Supplementary Fig. 1i, j). In addition, we also predicted the response elements of HNF4A in PRSS35 promoter (Supplementary Fig. 1k, upper panel), and further confirmed these response elements through luciferase assay. The results showed that all the four predicted HNF4A response elements (especially elements 1 and 3) in PRSS35 promoter were responsible for the transcription of PRSS35 (Supplementary Fig. 1k, lower panel). These data are consistent with the previous report that HNF4A had low expression in HCC[21,22]. Taken together, our data demonstrate that HNF4A regulates PRSS35 transcription in HCC.

### PRSS35 functions as an active protease

PRSS35 is annotated as an inactive pseudo serine protease due to the presence of a threonine at the canonical serine active site that defines this enzyme family[23]. However, recent studies have suggested that its function is associated with oocyte maturation, fertilization, and embryo development or tubulointerstitial inflammation in mouse kidney, even modulating wound-induced skin tumorigenesis[24–28]. Our observation of an association between decreased PRSS35 and HCC led us to hypothesize that this "pseudo protease" could show similar function as an active serine protease. To test this possibility, we first examined PRSS35 protease activity toward β-casein, a broad-spectrum substrate of many serine proteases[29]. To this end, we purified full-length PRSS35 with a fused His-tag (FL-PRSS35) from Escherichia coli, and then incubated it with β-casein. We observed that β-casein was completely cleaved in the presence of purified FL-PRSS35, suggesting that PRSS35 is an active protease (Fig. 2a, left panel). Since many proteases are known to undergo activation through cleavage by pro-protein convertases (PCs) before functioning[30,31], we conducted WB analysis of FL-PRSS35 in the in vitro reaction system. This analysis revealed the presence of multiple truncated forms of PRSS35 (Fig. 2a, right panel), suggesting that FL-PRSS35 function potentially required PC cleavage into short, active forms.

Additionally, we determined the PRSS35 cleavage site in β-casein through LC–MS analysis of full length and cleaved β-casein peptide sequence. The results showed that peptide fragments from full-length β-casein, but not cleaved β-casein, contained amino acid residues immediately in front of K44 (Fig. 2b and Supplementary Fig. 2a), which indicated that the position between K43 and K44 served as the cleavage site for PRSS35.

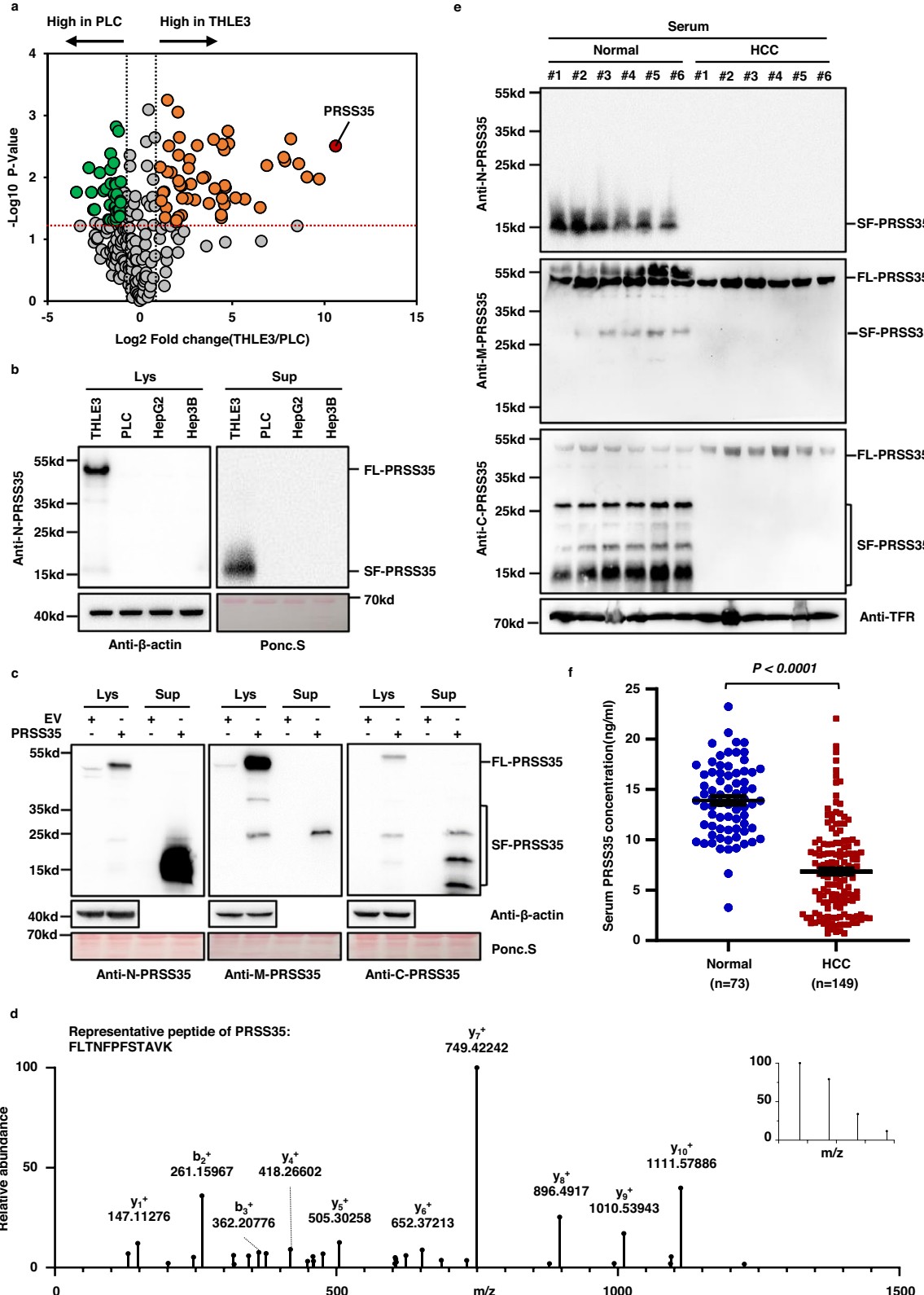

## PRSS35 is activated by proprotein convertase

To investigate whether PRSS35 is cleaved by PCs to produce the active mature forms, we studied the effects of FURIN, a well-known and ubiquitous PC, on PRSS35 cleavage. We generated FURIN over-expression and knockdown PLC cell lines, then used WB to determine FL-PRSS35 and SF-PRSS35 levels in each line. These experiments showed that FL-PRSS35 levels decreased while SF-PRSS35 increased

under FURIN overexpression, whereas the opposite effects were observed when in the FURIN knockdown lines (Fig. 2c), indicating that PRSS35 was cleaved by FURIN and potentially other PCs. Bioinformatics analysis of PRSS35 protein sequence revealed at least six potential PC cleavage sites in *human* PRSS35[32–34]. We further predicted the PC cleavage sites of PRSS35 from other species and identified three cleavage sites that were well conserved across all species represented

**Fig. 1 | PRSS35 is a secreted protein that decreased in HCC patients. a** Proteomic analysis and comparison of THLE3 and PLC secretome (without non-classical secreted proteins) presented as a volcano plot. The red transverse dashed line indicates adjusted *P*-value of 0.05. The left black longitudinal dashed line indicates a fold-change (FC) of 0.5 and the right black longitudinal dashed line indicates a FC of 2.0. Orange dots: significantly increased proteins in THLE3 secretome (*P* < 0.05, FC > 2.0). Green dots: significantly decreased proteins in THLE3 (*P* < 0.05, FC < 0.5). The experiment was repeated three times. **b** Western blot analysis of intracellular and extracellular PRSS35 protein levels using anti-N-PRSS35 antibody in THLE3, PLC, HepG2 and Hep3B cells. Ponceau staining and β-actin served as loading control. FL full length, SF short form, Lys lysate (intracellular proteins), Sup supernatant (extracellular proteins). **c** Western blot analysis of intracellular and extracellular PRSS35 protein levels using three different antibodies in PLC cells stably expressing PRSS35 or control empty vector (EV). Ponceau staining and β-

actin served as loading control. FL full length, SF short form. **d** Collection of extracellular proteins from PLC cells stably expressing PRSS35, followed by SDS-PAGE analysis and mass spectrometry analysis of extracellular PRSS35 protein in SDS-PAGE gel where SF-PRSS35 located in. Representative PRSS35 peptide from MS/MS spectra was shown. The inset of the MS/MS spectrum was the precursor's MS spectrum. **e** PRSS35 protein levels were determined with three different antibodies by western blot using the serum samples from normal subjects (Normal) and human HCC patients (HCC), respectively. Transferrin (TFR) served as loading control. **f** Serum PRSS35 concentration was measured by customized ELISA kit from 73 normal subjects and 149 human HCC patients. Data are presented as the mean ± s.e.m. (**f**). Statistical significance was determined by two-tailed unpaired Student's *t*-test (**f**). The blotting experiments were repeated at least three times with biological replicates (**b**, **c**, **e**). Source data are provided as a Source Data file.

in the UniProt database, including mouse and rat (Fig. 2d and Supplementary Fig. 2b). Notably, the three additional predicted cleavage sites were present only in *human* PRSS35 in the region near site 2, which we designated the PC cleavage site enriched region (PCSER; Fig. 2d).

To confirm that these three highly conserved, predicted cleavage sites in PRSS35 were bona fide PC recognition sites, we introduced non-synonymous mutations (AAAA) to each site individually (Fig. 2d). These mutations disrupted PRSS35 cleavage, with each PRSS35 mutant showing different, specific band patterns in WB analysis (Fig. 2e). First, the mutation at site 3 (M3) resulted in the disappearance of all SF-PRSS35, suggesting that M3 was essential for all PRSS35 cleavage, meaning cleavage at this site was required for subsequent proteolytic processing (Fig. 2e, compare lane 4 with lane 1). Mutation at site 2 (M2) resulted in decreased accumulation of short forms consisting of domain1 (D1, 12.4kd) or domain3 (D3, 14.2kd). Moreover, the intermediate fragment consisting of domains 1, 2, 3, and the PCSER (i.e., D1 + D2 + PCSER + D3) increased (Fig. 2e, compare lane 3 with lane 1), which suggested that cleavage at site 2 facilitated subsequent cleavage into short forms of PRSS35. In contrast, site 1 mutation (M1) only suppressed cleavage leading to production of domain1 (D1, 12.4kd), and resulted in the accumulation of an intermediate fragment carrying domains 1 and 2 (D1 + D2) (Fig. 2e, compare lane 2 with lane 1). These findings suggested that cleavage sites we predicted are real PC recognition sites on PRSS35 and cleavage of these sites are essential for the maturation of PRSS35.

In order to determine which form(s) of PRSS35 exhibit active protease functions, we then purified His-D1, Flag-D2, and Myc-D3 fusion protein fragments from *E. coli*. Incubation of each of these purified protein fragments with β-casein showed that β-casein was only cleaved in the presence of D1 in vitro (Fig. 2f and Supplementary Fig. 2c), suggesting that only the D1 short form is active. Taking into account the molecular weight of the short form presented in the in vitro reaction, our data consolidate that it's the D1 short from (12.4 kD), but not the FL-PRSS35, that cleaved β-casein (Fig. 2a). Taken together, these results demonstrated that PRSS35 functions as a protease following cleavage by a PC to produce its active, truncated form carrying only domain1 (D1).

To better understand the working model and different roles of these predicted cleavage sites in PRSS35 maturation, we performed further bioinformatics analysis[35] and found that the 16 amino acids in the N terminal of PRSS35 were likely to be the signal peptide (Supplementary Fig. 2d). In light of our WB data, we therefore proposed that an initial cleavage site in the N-terminus removed the signal peptide[36] (SP) from FL-PRSS35 (Fig. 2g, conformation A), resulting in a folded conformation of PRSS35 in which site 3 was exposed at the protein surface and sites 1 and 2 were buried in the protein interior (Fig. 2g, conformation B). Cleavage at site 3 by PCs then changed the conformation of PRSS35 to expose sites 1 and 2. Then, PCs could be recruited to site 2 in the adjacent PCSER and site 1 (Fig. 2g,

conformation C), ultimately resulting in complete cleavage of PRSS35 into an active D1-containing short form, as well as inactive fragments harboring D2, D3 (Fig. 2g, conformation D).

To further identify the cleavage motif (s) recognized by PRSS35, and consequently its target substrates, we employed a high-throughput protease screen (HTPS), as described in other studies[37–39]. This analysis identified 1318 cleavage windows for PRSS35 in 22 potential substrates (Supplementary Data 1). Further analysis of the cleavage windows revealed that two adjacent lysine amino acids (KK) served as the core cleavage motif recognized by PRSS35 (Fig. 2h), which was consistent with the cleavage site we identified in β-casein (Fig. 2b). In addition, we synthesized three fluorogenic peptides containing the predicted cleavage motif (Fig. 2h and Supplementary Fig. 2e, right panel), and found that the His-D1 peptide fragment could cleave all three fluorogenic peptides in vitro (Supplementary Fig. 2e, left panel). Because PRSS35 cleaves substrates at the di-basic KK residues as a trypsin like serine protease, we co-incubated β-casein and PRSS35-Domain1 in the presence of common serine protease inhibitors and observed whether it led to the loss of PRSS35 activity. The results showed that PRSS35 lost the ability to cleave β-casein in the presence of serine protease inhibitors, PMSF or protease inhibitor cocktail (Supplementary Fig. 2f). These data proved that PRSS35 is an active trypsin like serine protease. Thus, the His-Asp-Ser triad should be the structural basis for the PRSS35 protease activity[40]. Further analysis of PRSS35-Domain1 sequence revealed the potential catalytic sites in PRSS35 based on His-Asp-Ser triad (Data not shown). Importantly, mutation of each serine residue in those sites demonstrated that serine 117 was the catalytic site of PRSS35 (Supplementary Fig. 2g). Collectively, these findings demonstrated that PRSS35 is a trypsin like serine protease activated by PCs cleavage, and that its activated short form, D1, can degrade substrates through targeting a KK recognition site.

## PRSS35 inhibits HCC development in vivo

Our data showed that PRSS35 is an active protease that is suppressed in HCC, which led us to investigate the relationship between PRSS35 and tumorigenesis. We found that PRSS35 overexpression had no effect on the proliferation of cultured liver cancer cells, including PLC, HepG2, Hep3B and Hepa1-6 cells (Supplementary Fig. 3a). However, overexpression of *mouse* PRSS35 (mPRSS35) significantly suppressed growth of Hepa1-6 cells in C57BL/6J mice, compared to that in the EV control group (Fig. 3a and Supplementary Fig. 3b). We then established a spontaneous HCC mouse model through hydrodynamic injection of plasmid YAP-5SA[41]. The results showed that mPRSS35 overexpression markedly suppressed development of liver cancer in mice (Fig. 3b and Supplementary Fig. 3c). To further validate these observations, we generated *mPRSS35* knockout mice (*mPRSS35*-KO) using Crisper/Cas9 base editing (Supplementary Fig. 3d), and found that the *PRSS35*-KO mice were healthy and had the similar physical conditions as the WT mice (Supplementary Fig. 3e–j). In the YAP-5SA-induced-HCC mouse

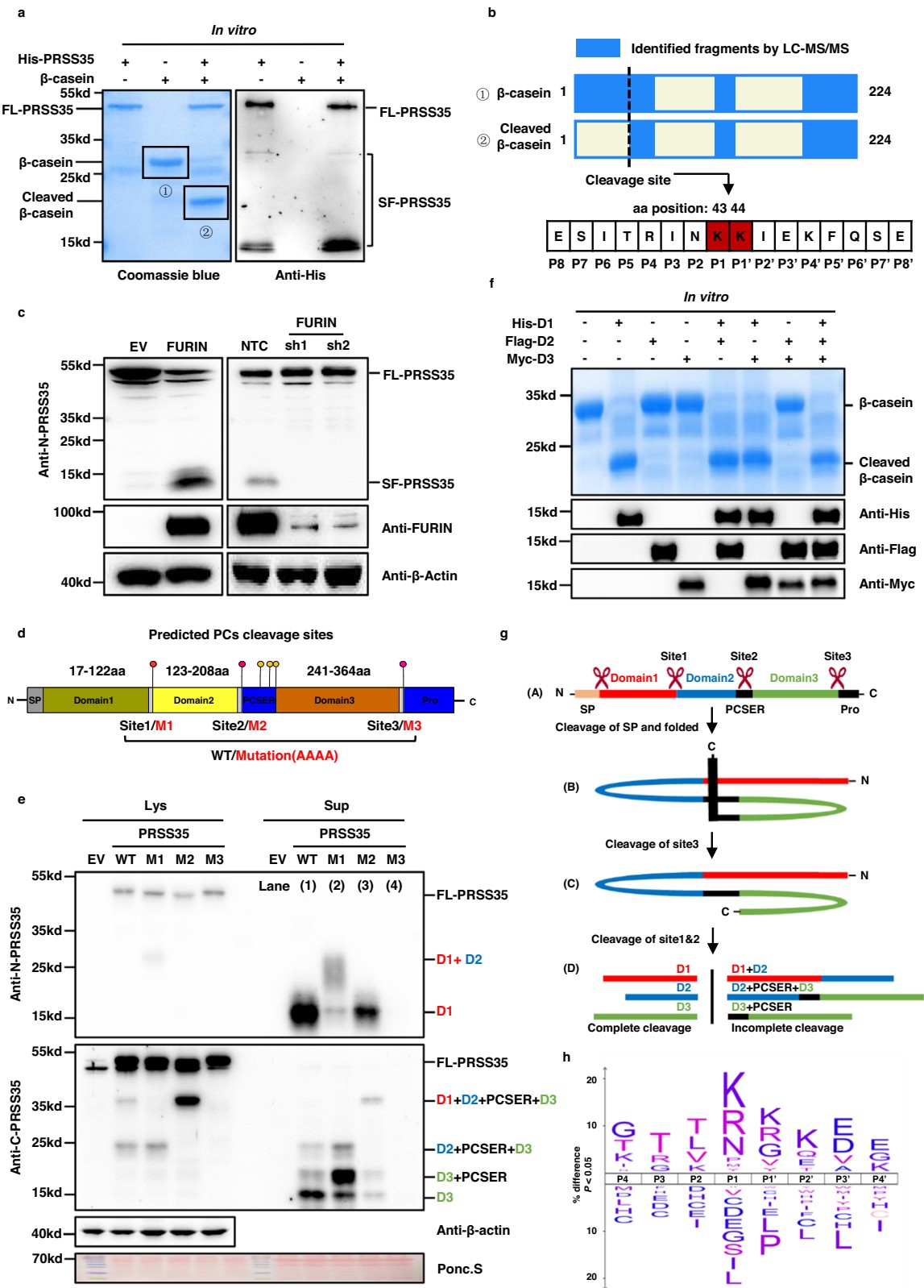

model, we observed that *mPRSS35*-KO mice developed liver cancer at an accelerated rate compared to WT mice (Fig. 3c and Supplementary Fig. 3k). In addition to the Hepa1-6 murine HCC model and YAP-5SA-induced-HCC murine model, we also confirmed the role of PRSS35 on tumorigenesis in vivo with a xenograft HCC model. We found that *human*-PRSS35 (hPRSS35) overexpression significantly suppressed HepG2 tumor growth in Balb/c-nude mice (Fig. 3d). These results

demonstrated that mPRSS35 contributed to tumor suppression in mice, albeit its lack of effects on HCC cell growth in vitro.

## CXCL2 is the substrate of PRSS35
Since PRSS35 only exhibited tumor-suppressive effects in mice, but not in the cultured cells, we speculated that some factors in the microenvironment are affected by PRSS35. To explore this possibility,

**Fig. 2 | PRSS35 activated by FURIN functions as a protease. a** His-PRSS35 protein purified from *E. coli* was incubated with β-casein protein at 37 °C overnight, followed by SDS-PAGE and coomassie brilliant blue staining (left panel). PRSS35 signal was determined by western blot with anti-His antibody (right panel). FL full length, SF short form. **b** Bands from (**a**) were analyzed by mass spectrometry to identify PRSS35 cleavage sites. Schematic of β-casein coverage by LC–MS/MS identified peptides and schematic of PRSS35 cleavage site in β-casein (upper panel). PRSS35 cleavage of β-casein at K43-K44 was monitored by LC–MS/MS (lower panel). **c** Western blot analysis of PRSS35 protein levels in HepG2 cells stably expressing FURIN or EV (left panel) or in HepG2 cells stably expressing shFURIN or NTC (right panel). β-actin served as loading control. FL full length, SF short form. **d** Schematic diagram of the PRSS35 protein and the predicted proprotein convertases (PCs) cleavage sites in PRSS35. Conserved sequences of proprotein convertases cleavage sites in PRSS35 and corresponding mutant sequences are shown. PCSER: PCs cleavage sites enriched region. **e** Western blot analysis of intracellular and extracellular PRSS35 protein levels with anti-N-PRSS35 and anti-C-PRSS35 antibodies using PLC cells expressing PRSS35-WT, PRSS35-M1, PRSS35-M2, PRSS35-M3 or EV. Lys lysate (intracellular proteins), Sup supernatant (extracellular proteins), FL full length, PCSER PCs cleavage sites enriched region. D1: PRSS35 domain1. D2: PRSS35 domain2. D3: PRSS35 domain3. **f** *E. coli* purified His-D1, Flag-D2 and Myc-D3 protein alone, or in combination, were incubated with β-casein protein at 37 °C overnight, followed by SDS-PAGE and coomassie brilliant blue staining. His-D1, Flag-D2 and Myc-D3 signals were determined by western blot with anti-His, anti-Flag and anti-Myc antibodies. FL full length, SF short form, D1 PRSS35 domain1, D2 PRSS35 domain2, D3 PRSS35 domain3. **g** The working model depicts the sequential self-cleavage of PRSS35 protein. **h** An iceLogo (upper panel) generated from 1318 purified human FL-PRSS35 and PRSS35-domain1 cleavage sites identified from PLC, HepG2, Hep3B and 293T cells with the human Swiss-Prot proteome as reference set (lower panel). In this logo, PRSS35 cleavage occurred at the peptide bond between residues position1 (P1) and position1' (P1'). The blotting experiments were repeated at least three times with biological replicates (**a**, **c**, **f**, **e**). Source data are provided as a Source Data file.

we performed SILAC proteomic analysis in conjunction with RNA sequencing to identify secreted proteins that decreased in accumulation, but not in transcription level, in the culture medium of PRSS35 overexpressing PLC cells. This analysis identified nine secreted proteins as potential substrates of PRSS35 (Fig. 3e and Supplementary Fig. 3l). Since the tumor-suppressive activity of PRSS35 appeared to depend on the in vivo microenvironment, we focused our study on CXCL2 as a candidate substrate, given its well-established role as a chemokine in the tumor immune microenvironment[42–44].

WB analysis indicated that CXCL2 was mainly detected as a secreted protein in culture medium. More importantly, overexpression of PRSS35 markedly decreased the extracellular levels of CXCL2 in HepG2, PLC and Hep3B cells (Fig. 3f). Furthermore, peptide sequence analysis showed that CXCL2 harbored the KK cleavage site (Fig. 3g, right panel), and mutation of this motif blocked CXCL2 cleavage by PRSS35 (Fig. 3g, left panel), further supporting that CXCL2 was a substrate of PRSS35. To exclude other indirect mechanisms by which PRSS35 might decrease CXCL2 protein levels, recombinant D1 purified from *E. coli* was incubated with *E. coli* purified CXCL2 protein, resulting in CXCL2 degradation (Fig. 3h), which confirmed CXCL2 as a direct substrate of PRSS35 proteolytic activity in vitro.

Since CXCL2 is a chemokine that functions in recruiting neutrophils[42–45], we therefore sought to determine whether the CXCL2 degradation by PRSS35 could suppress the recruitment of neutrophils. For this purpose, we isolated neutrophils from the peripheral blood of mice (Supplementary Fig. 3m) and treated them with conditional media from Hepa1-6 cells expressing EV, Flag-mPRSS35, mCXCL2, or both. Analysis of the neutrophil migration[46,47] showed that the conditioned medium collected from mCXCL2-expressing cells enhanced neutrophil migration, which was abolished by overexpression of mPRSS35 (Fig. 3i, upper panel). Consistent with these findings, WB indicated that mCXCL2 protein in conditioned medium was significantly degraded in the presence of mPRSS35 (Fig. 3i, lower panel). Similar results were observed for the active D1 form of PRSS35 (Fig. 3j), thereby demonstrating that mPRSS35-mediated degradation of mCXCL2 was a contributing factor in neutrophil activity.

## PRSS35 suppresses HCC progression via decreased neutrophil recruitment to tumors and attenuated NETs formation

In order to investigate the effects of PRSS35 degradation of CXCL2 on neutrophil activity in tumor progression in vivo, we generated Hepa1-6 cell lines that stably expressed mPRSS35, mCXCL2, both, or an EV control and separately inoculated these lines subcutaneously into C57BL/6J mice. The results indicated that mCXCL2 overexpression promoted tumor growth in mice, while mice expressing mPRSS35 showed the lowest tumor development, with or without mCXCL2 co-expression (Fig. 4a and Supplementary Fig. 4a). WB analysis of tumor tissues confirmed that mCXCL2 protein was depleted in the presence of mPRSS35 (Fig. 4b). Further examination of the tumor tissues by flow cytometry and IHC revealed enhanced neutrophil recruitment into tumors in mice overexpressing mCXCL2, which was abolished by co-expression with mPRSS35 (Fig. 4c, e), indicating that mPRSS35-mediated mCXCL2 degradation can limit neutrophil recruitment in HCC tissues in mice.

As previously reported, neutrophils promote tumor progression via upregulation of NETs[48–50]. We therefore examined the levels of citrullinated histone H3 (H3Cit), a marker for NETs formation, in tumor tissues using WB, and found it was indeed elevated in mice overexpressing mCXCL2, but attenuated under mPRSS35 overexpression (Fig. 4b). These results strongly suggested that PRSS35 and CXCL2 induced opposite effects in the formation of NETs. In addition, we detected H3Cit and another indicator of NETs formation, the myeloperoxidase-DNA (MPO−DNA) complex, in the serum of tumor-bearing mice by ELISA[51,52]. Consistent with mCXCL2 promotion of H3Cit levels, we found that H3Cit and MPO−DNA complex levels were markedly increased in the serum of mice harboring tumors from mCXCL2 overexpressing Hepa1-6 cells, but not in mice harboring tumors from mPRSS35 overexpressing Hepa1-6 cells with or without mCXCL2 overexpression (Fig. 4d, Supplementary Fig. 4f, left panel), thereby demonstrating that PRSS35 can downregulate NETs formation by suppressing CXCL2-mediated neutrophil recruitment to tumors in vivo.

In YAP-5SA-induced-HCC model mice, consistent with the above results obtained in transgenic Hepa1-6-induced-HCC mice, we found that tumor development was accelerated in *mPRSS35*-KO mice compared to that in WT mice, while shRNA knockdown of mCXCL2 (shmCXCL2) attenuated this effect (Fig. 4f and Supplementary Fig. 4b). Moreover, we observed higher neutrophil infiltration in tumors and NETs formation in both tumor tissues and mouse serum in tumor-bearing *mPRSS35*-KO mice, but were inhibited in the presence of shmCXCL2 (Fig. 4g and Supplementary Fig. 4c−e). WB analysis confirmed the inhibitory effects of PRSS35 on CXCL2 protein levels in tumor tissues (Supplementary Fig. 4d). Collectively, these data demonstrated that PRSS35 could inhibit neutrophil infiltration and NETs formation in tumors via degradation of CXCL2.

To further study whether neutrophils and NETs are indeed involved in PRSS35-mediated tumor suppression, Hepa1-6 cells stably expressing empty vector (EV) or Flag-mPRSS35 were injected subcutaneously into C57BL/6J mice. At nine days after cell injection, we injected the neutrophil neutralizing Ly6G antibody, or IgG control, into tumors and surrounding subcutaneous tissues in three-day intervals[53]. Again, mPRSS35 expression significantly suppressed tumor growth compared to that in the EV control group, while treatment with Ly6G led to similar effects in tumors with or without Flag-mPRSS35 overexpression (Fig. 4h and Supplementary Fig. 4g). Consistently, mPRSS35 expression significantly suppressed neutrophil tumor

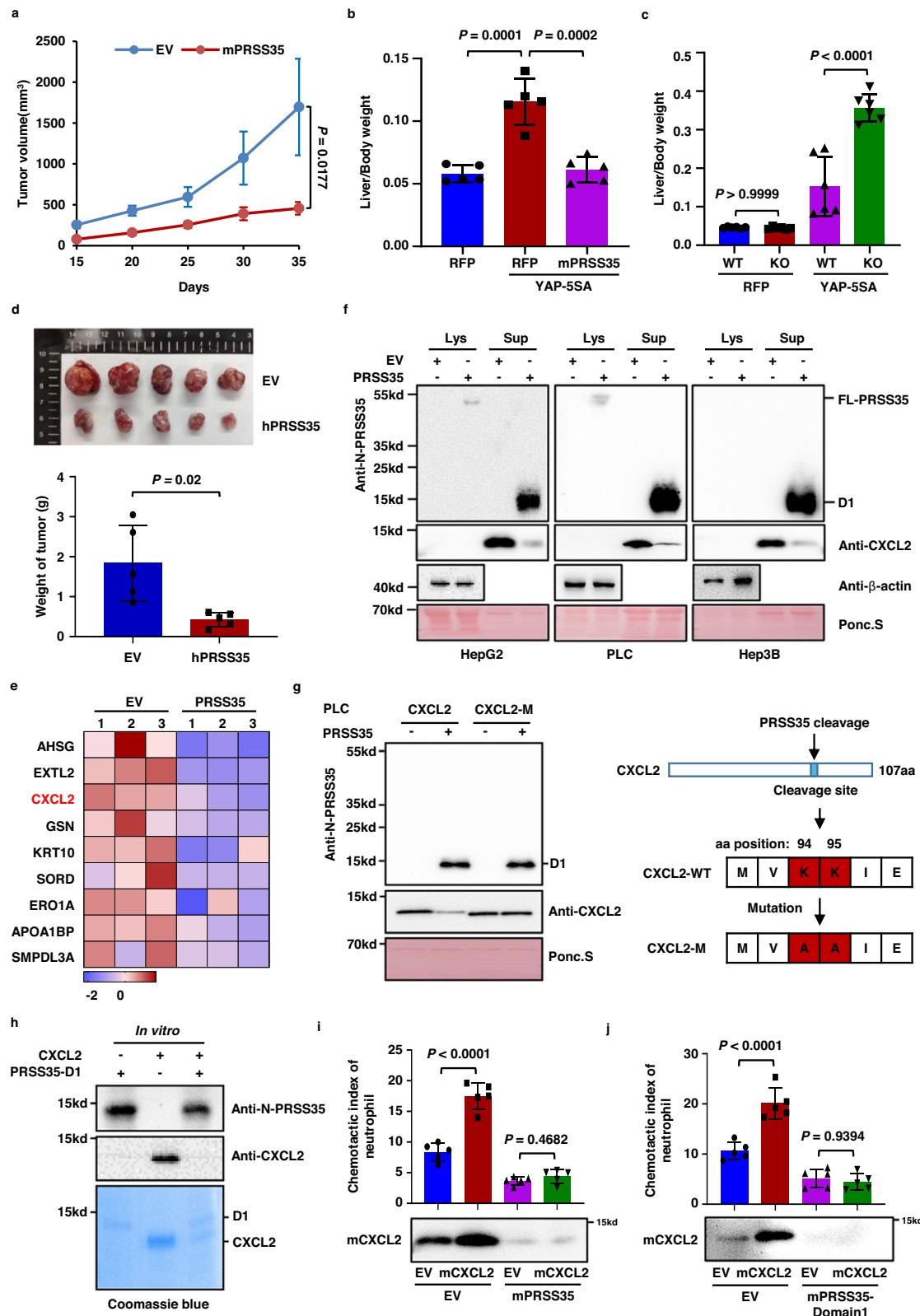

infiltration and NETs formation, and Ly6G treatment resulted in similar effects in tumors with or without Flag-mPRSS35 overexpression (Fig. 4i–k and Supplementary Fig. 4f, right panel, 4h, i).

To determine whether the inhibition of tumor growth by PRSS35 is due to NETosis, we also treated the mice with DNase I rather than anti-Ly6G antibody and replicated the experiments performed in Fig. 4h–k. The results showed that, similar as mPRSS35 overexpression, NETs

elimination by DNase I treatment significantly suppressed the tumor growth (Supplementary Fig. 4j). The infiltration of neutrophils was not affected by DNase I, but was suppressed by mPRSS35 (Supplementary Fig. 4l, m). Importantly, NETs were significantly decreased by both mPRSS35 overexpression and DNase I treatment in mouse serum as well as in tumor tissue samples (Supplementary Fig. 4k, n). These results thus indicated that PRSS35 functions as an upstream regulator of

**Fig. 3 | PRSS35 suppresses neutrophil migration by degrading CXCL2. a** Hepa1-6 cells stably expressing Flag-mPRSS35 or EV were injected subcutaneously into C57BL/6J mice. Tumor growth was measured starting from 15 days after inoculation (*n* = 6 biological replicates). **b** Plasmids expressing YAP-5SA alone or YAP-5SA plus Flag-mPRSS35 together with PB transposase plasmids were delivered into ICR (Institute of Cancer Research) mice by hydrodynamic injection. YAP-5SA induced liver tumorigenesis was analyzed approximately 100 days after injection. Red fluorescent protein (RFP) served as a control. Liver/body weight ratios were measured at the end of the experiment (*n* = 5 biological replicates). **c** Plasmids expressing YAP-5SA or RFP together with PB transposase plasmids were delivered into *mPRSS35*-KO or WT C57BL/6J mice by hydrodynamic injection. YAP-5SA induced liver tumorigenesis was analyzed approximately 100 days after injection. Liver/body weight ratios were measured at the end of the experiment (*n* = 6 biological replicates). **d** Equal numbers of HepG2 cells overexpressing EV or hPRSS35 were injected subcutaneously into Balb/c-nude mice. Photograph and weight of tumors at the end of the experiment (35 days after injection, *n* = 5 biological replicates). **e** Heat map analysis of SILAC secretomics data. The proteins significantly decreased in PLC cells stably expressing PRSS35 compared to EV were shown. Red indicates relative high level, whereas blue indicates relative low level.

**f** Western blot analysis of intracellular and extracellular CXCL2 protein levels in HepG2, PLC and Hep3B cells stably expressing PRSS35 or EV. Ponceau staining and β-actin served as loading control. **g** Western blot analysis of the cleavage of CXCL2 by PRSS35 after mutation of K94 and K95 into A94 and A95 in PLC cells (left panel). Diagram shows mutation site of CXCL2 (right panel). **h** PRSS35-D1 protein purified from *E. coli* was incubated with CXCL2 protein at 37 °C for 48 h. D1: PRSS35 domain1. **i** Neutrophils migration rates were determined in the conditional medium from Hepa1-6 stably expressing EV, mCXCL2, Flag-mPRSS35, or mCXCL2 plus Flag-mPRSS35 (upper panel, *n* = 5 biological replicates). Western blot analysis of mCXCL2 protein levels in Hepa1-6 cells stably expressing EV, mCXCL2, Flag-mPRSS35, or mCXCL2 plus Flag-mPRSS35 (lower panel). **j** Neutrophils migration rates were determined in the conditional medium from Hepa1-6 stably expressing EV, mCXCL2 or their medium treated with purified mPRSS35-domain1 (upper panel, *n* = 5 biological replicates). Western blot analysis of mCXCL2 protein levels in above conditional medium (lower panel). Data are presented as the mean ± s.d. (**a–d**, **i**, **j**). Statistical significance was determined by two-way ANOVA (**a–d**, **i**, **j**). The blotting experiments were repeated at least three times with biological replicates (**f**, **g–j**). Source data are provided as a Source Data file.

neutrophil activity in tumor suppression in vivo. Our cumulative results demonstrate that PRSS35 suppressed HCC progression by inhibiting CXCL2-mediated neutrophil recruitment and NETs formation.

## Discussion

A well-defined hallmark of cancers is the reprogramming of gene expression to promote cancer cell survival and proliferation. Historically, cancer research has been focused generally on abnormal gene function in individual cancer cells such as exploring the effects and mechanisms of genome instability, sustained proliferative signaling, resistance to cell death, and metabolic reprogramming[2,54–58]. However, the aberrant expression of secreted proteins, which function outside of cancer cells and serve as a major route of communication between cells, remains largely overlooked. In this study, we identify a secreted tumor suppressing protein, PRSS35, and describe its function in controlling neutrophil recruitment and cancer progression. Mechanistically, we show that PRSS35 is a potent secreted protease activated by PC cleavage and, through its substrate CXCL2, PRSS35 suppresses HCC progression in vivo via inhibition of CXCL2-mediated neutrophil recruitment and NETs formation (Supplementary Fig. 4p).

Recent studies of quantifying neutrophil infiltration into the tumor microenvironment (TME) have revealed their diversity and plasticity in cancer development[10]. In general, neutrophils could play dual roles in tumor progression due to their diversity and plasticity. On one hand, neutrophils could release ROS, miR-155, and miR-23A to induce genetic instability, thereby initiating tumor[59–61]. Neutrophils could also produce some secreted factors, such as epidermal growth factor (EGF), hepatocyte growth factor (HGF) and platelet-derived growth factor (PDGF) to sustain tumor proliferation[62,63]. Furthermore, neutrophils could also promote tumor angiogenesis through the secretion of the angiogenesis factors (BV8, S100A8, S100A9), and even MMP9, a protease that could cleave and then activate vascular endothelial growth factor A (VEGFA)[64–67]. In immunity regulation, neutrophils produce arginase 1 (ARG1) to inhibit CD8+ T cell expansion, thereby preventing antitumor immune responses in mice[68]. More recently, Zhou et al. also reported tumor-associated neutrophils could recruit macrophages and T-reg cells to promote progression of HCC and even resistance to sorafenib, a clinical drug for liver cancer[53]. Hangai et al. also reported that accumulation of neutrophils in tumor could limit CD8+ T and NK cell numbers and antitumor activity[69]. On the other hand, in early non-small-cell lung cancer (NSCLC), neutrophils which express OX40 ligand, CD86 and 4-1BB ligand could assist the activation of CD4+ T cells to defense tumor cells[70,71]. Neutrophils could also kill tumor cells directly through cell to cell contact and the production of ROS[72–74]. Furthermore, neutrophils promoted the production of IL-12 of macrophages, and then promoted polarization of the UTCαβ to increase the production of IFNγ

to establish effective antitumor immunity during the early phase of sarcoma[75]. In general, neutrophils could play both pro-tumor and anti-tumor roles due to their various plasticity in response to multiple TME stimulation.

As described above, neutrophils can affect cancer development through multiple mechanisms, among which, emerging evidence establishes NETs formation as a major determining factor in tumor progression[13]. As reported previously, NETs play an important role in promoting tumor growth through directly affecting tumor cells or modifying immunity microenvironment. Recent reports have revealed that NETs drive mitochondrial homeostasis in tumor cells, directly augmenting tumor growth. Mechanistically, neutrophil elastase (NE) released from NETs activated TLR4 on tumor cells, leading to PGC1a upregulation, increased mitochondrial biogenesis, and accelerated tumor cell growth[76]. NETs could also impair tumor cell contact with cytotoxic immune cells, leading to evasion of immunological surveillance[47]. Interestingly, we observed in this study that decreased PRSS35 levels led to an increase in the chemokine CXCL2, resulting in greater neutrophil accumulation and increased NETs formation in the TME. Our study thus provides strong evidence linking the secreted protease PRSS35 with neutrophil accumulation and NETs formation in tumors, and expands our understanding of the regulatory mechanisms controlling neutrophil recruitment and activity in the TME.

As a malignant tumor, the accurate diagnosis of early-stage HCC remains challenging, which contributes to the high mortality of the affected patients[77]. Thus, the identification of reliable biomarkers represents an essential step in reducing mortality among HCC patients. However, the progress of our endeavors has been hampered by the complex nature of body fluid samples, especially for serum, and the large dynamic range between the concentrations of different proteins, which led to considerable gaps persisting in the development of biomarkers and our understanding of the scope and mechanisms of secreted proteins that participate in cancer progression. In the current study, we employed a label-free proteomic analysis of the secretome of HCC cells and identified a close relationship between the secreted protein PRSS35, NETs and HCC prognosis, suggesting its potential value as a biomarker for screening early-stage HCC as well as a potential candidate target for HCC therapeutic strategies. Since CXCR1/2 axis is involved in NAFLD and mediates neutrophil function which contributed to the development of HCC in NAFLD[50,78–80], it would be interesting to confirm the relationship between PRSS35 and NAFLD. However, this warrants further independent studies.

## Methods

Our research complies with all relevant ethical regulations of the University of Science and Technology of China. All animal protocols

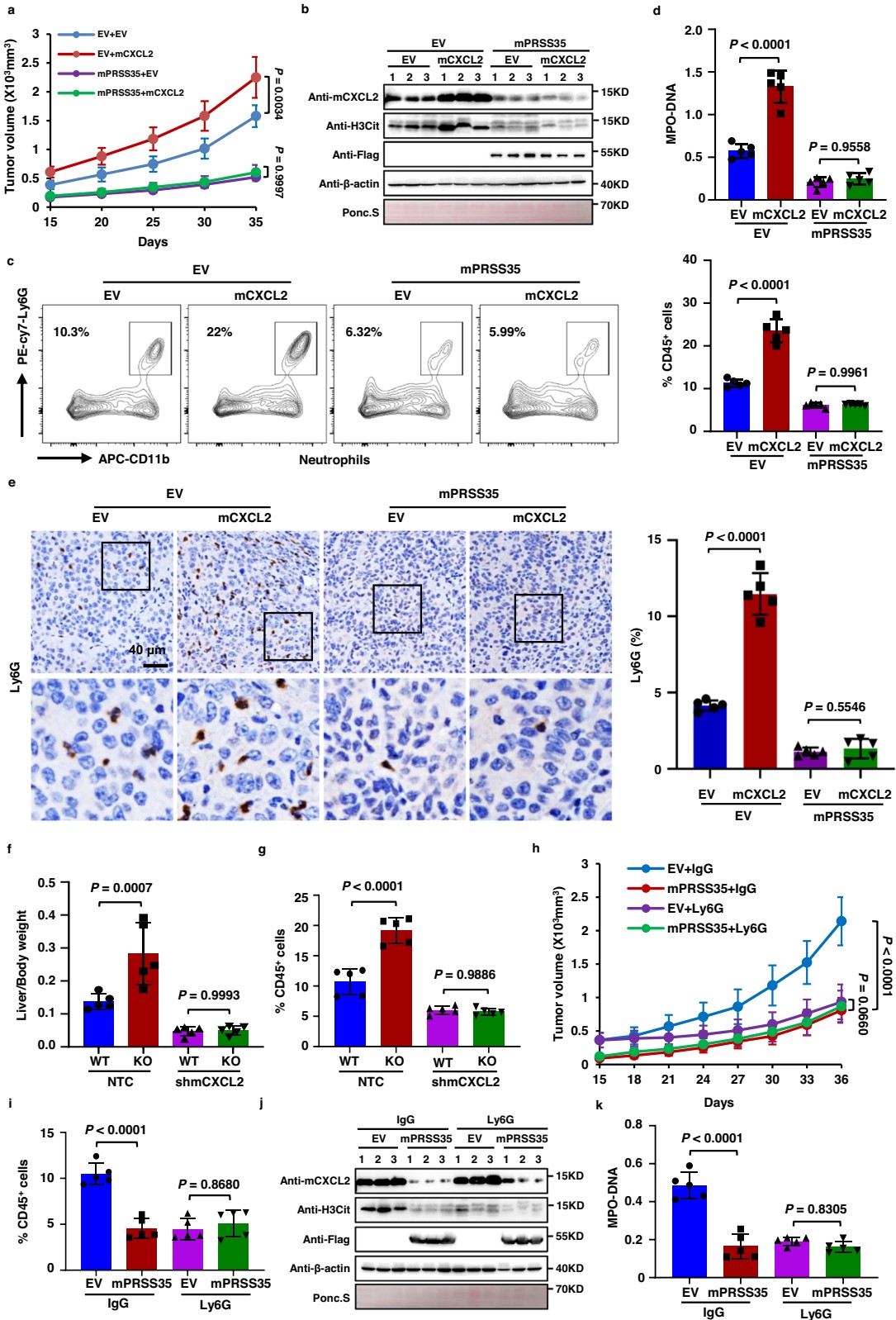

were approved by the Animal Research Ethics Committee of the University of Science and Technology of China and were performed following the guidelines for the use of laboratory animals. The collection and use of clinical materials were approved by the Institutional Research Ethics Committee of the First Affiliated Hospital of University of Science and Technology of China.

## Cell culture and reagents

Human HepG2, Hep3B, PLC, Hepa1-6 and HEK293T cells were cultured in DMEM containing 25 mM glucose, 4mM L-glutamine and 1 mM pyruvate (Gibco, 12800). THLE3 cells were cultured in BEGM (LONZA, CC3170). All cell lines were tested routinely to make sure they are negative for mycoplasma contamination by Mycoplasma PCR

**Fig. 4 | PRSS35 inhibits HCC development by suppressing CXCL2-mediated neutrophil NETs formation. a–e** Hepa1-6 cells stably expressing EV, mCXCL2, Flag-mPRSS35, or mCXCL2 plus Flag-mPRSS35 were injected subcutaneously into C57BL/6J mice. **a** Tumor growth was measured starting from 15 days after inoculation (n = 5 biological replicates). **b** Western blot analysis of mCXCL2, Flag-mPRSS5 and H3Cit protein levels in tumor lysates. Ponceau staining and β-actin served as a loading control. **c** Tumors were extracted at the end of the experiment. The frequency of neutrophil in tumors was detected by Flow cytometry (n = 5 biological replicates). **d** Serum NETs levels were measured by MPO–DNA ELISA kit. **e** Representative IHC images (left panel) and quantification (right panel) of neutrophil in the tumor (n = 5 biological replicates). **f, g** Plasmids expressing YAP-5SA alone or YAP-5SA plus mCXCL2 shRNAs together with plasmids expressing PB transposase were delivered into *mPRSS35*-KO or WT C57BL/6J mice by hydrodynamic injection. YAP-5SA induced liver tumorigenesis was analyzed approximately 100 days after injection. **f** Liver/body weight ratios were measured at the

end of the experiment (n = 5 biological replicates). **g** The frequency of neutrophil in livers was detected by Flow cytometry (n = 5 biological replicates). **h–k** Hepa1-6 cells stably expressing EV or Flag-mPRSS35 were injected subcutaneously into C57BL/6J mice, followed by intratumor and peritumoral injection of Ly6G antibody or IgG antibody nine days later. **h** Tumor growth was measured starting from 15 days after inoculation (n = 5 biological replicates). **i** Tumors were extracted at the end of the experiment. The frequency of neutrophil in tumors was detected by Flow cytometry (n = 5 biological replicates). **j** Western blot analysis of mCXCL2, Flag-tag and H3Cit proteins in tumor lysates. Ponceau staining and β-actin served as a loading control. **k** Serum NETs levels were measured by MPO–DNA ELISA kit (n = 5 biological replicates). Data are presented as the mean ± s.d. (**a, c–i, k**). Statistical significance was determined by two-way ANOVA (**a, c–i, k**). The blotting experiments were repeated at least three times with biological replicates (**b, j**). Source data are provided as a Source Data file.

detecting method. Cell line identities were confirmed by STR profiling. The medium was supplemented with 1% penicillin-streptomycin and 10% FBS. For serum starvation, cells were washed twice with PBS and cultured in the medium without FBS. For SILAC labeling, vehicle-expressing PLC cells and constructed *human* PRSS35-expressing PLC cells cultured in light DMEM containing $^{12}C_6$, $^{14}N_2$-Lys (K0) and $^{12}C_6$, $^{14}N_4$-Arg (R0) or heavy DMEM containing $^{13}C_6$, $^{15}N_2$-Lys (K8) and $^{13}C_6$, $^{15}N_4$-Arg (R10), respectively. Cells were cultured and passaged until the labeling efficiency reached 98%.

### Plasmids and established stable cells
All shRNAs in PLKO or pBL vector against *human FURIN* and *mouse CXCL2* were commercially purchased from Sigma-Aldrich. shRNA targeting sequences are listed in Supplementary Table 4. Coding sequences of *human PRSS35, CXCL2, FURIN* were subcloned into pCDH empty vector, respectively. Constructing three *human PRSS35*-mutations based on constructed pCDH-*PRSS35*. Constructing *human CXCL2*-mutation based on constructed pCDH-*CXCL2*. Mouse *PRSS35, CXCL2* was subcloned into pBL lentiviral vector. Three domains and full-length of *PRSS35* was also subcloned into pET22b with His-tag, Myc-tag or Flag-tag for subsequent in vitro purification. PLKO and pCDH with or without exogenous sequence were then co-transfected with plasmids encoding *VSVG* and *Δ8.9* into HEK293T packaging cells using PEI (Polysciences). PLC, HepG2, Hep3B or Hepa1-6 cells were infected with lentivirus containing polybrene and selected with 0.5 μg/ml puromycin to establish stable cells.

### Collection of secreted proteins mixture
Before harvesting secreted proteins, cells were cultured in serum starvation condition for 24 h. Collected the media and filtered with 0.22 μm filter (Millipore) to ensure removal of any dead cells. Thereafter samples were transferred to an Amicon Ultra-4 3 K molecular weight cut-off centrifugal filter (Millipore), spun down until the remaining liquid is up to minimum volume. Transferred the remaining liquid to new tubes as the concentrated secreted proteins mixture.

### Western blot and antibodies
For intracellular and membrane proteins, cells or tissues were harvested and total cellular protein was isolated using RIPA buffer (50 mM Tris-HCl, pH 8.0, 150 mM NaCl, 5 mM EDTA, 0.1% SDS, and 1% NP-40) supplemented with protease inhibitor cocktails (Roche). For secreted proteins, diluted the concentrated secreted proteins with isopyknic RIPA buffer supplemented with protease inhibitor cocktails (Roche). Protein concentration was measured using the Bradford assay kit (Sangon biology). Equal amount of proteins were loaded and separated by SDS-polyacrylamide gel electrophoresis (SDS-PAGE). β-actin or ponceaux served as loading control. The primary antibodies for immunoblotting were directed against: PRSS35 (three types of antibodies recognized three different peptide

sequences were customized in Abclonal or purchased from Thermo Fisher Scientific), *human* and *mouse* CXCL2 (Proteintech), FURIN (Proteintech), HA-tag (Proteintech), Myc-tag (Proteintech), Flag-tag (Proteintech), His-tag (Proteintech), β-actin (Proteintech), Calnexin (Proteintech), H3Cit (Abcam).

### Detection of serum PRSS35 by ELISA
PRSS35 serum concentrations were determined using a customized ELISA kit from Ray-bio. Briefly, 100 μl plasma were loaded in duplicates and PRSS35 abundance was determined. The recognized peptide sequence of ELISA kit was illustrated in Supplementary Fig. 1g. Data on all 149 patients and 73 donors are provided in Supplementary Table 5.

### In-Gel mass spectrometry
Gel core containing the protein band of interest from the polyacrylamide gel was cut into 1–1.5 mm³ pieces and placed in a 1.5 ml tube. The tube was spun to collect the gel pieces at the bottom of the tube. Protein sample in gel were reduced with 5 mM dithiothreitol (DTT) at 56 °C for 30 min and alkylated with 15 mM iodoacetamide (IAA) in the dark at room temperature for 30 min. The alkylation was quenched by DTT at the final concentration of 20 mM at room temperature. Wash gel pieces with 50 mM $NH_4HCO_3$ and discard the liquid with pipet. Add 30% acetonitrile (ACN) in 50 mM $NH_4HCO_3$ and vortex for 30 s. Add 50% ACN in 50 mM $NH_4HCO_3$. And discard the liquid. Then add 100% ACN and discard the liquid. Speed-vacuum for 10–30 min until gel pieces are dry. Add 50 mM $NH_4HCO_3$ containing trypsin to the dry gel pieces and incubate at 37 °C overnight. Remove and retain the solution in a fresh tube. Add 60% ACN with 5% trifluoroacetic acid (TFA) to the gel pieces and sonicate in the water bath 1 min. Combine it with the solution collected before and dry down the combined solution. Suspend the pellet in 0.1% formic acid and ready for mass spectrometry.

For the HPLC-MS/MS analysis, the peptides were analyzed on an QE-plus mass spectrometer (Thermo Fisher Scientific, Waltham, MA, USA) equipped with an EASY-nLC 1200 nano liquid chromatography (HPLC) system (Thermo Fisher Scientific, Waltham, MA, USA). The peptides were dissolved in mobile phase buffer A (0.1% formic acid (FA) in water) and loaded onto Precolumn (Nano Trap 100 μm i.d. × 20 mm, Acclaim PepMap100 C18, 5 μm, 100 Å) and Analytical column (PepMap C18 2 μm 50 μm × 150 mm NV FS 1200 bar) by an autosampler. Then the peptides were separated and eluted with a linear gradient of 3–8% of mobile phase buffer B (0.1% FA in 80% acetonitrile) for 3 min, 8–28% buffer B for 70 min, 28–45% buffer B for 35 min, 45–90% buffer B for 2 min, and 90% buffer B for 10 min at the flow rate of 300 nL/min on HPLC system. The eluted peptides were ionized under high voltage (2.2 KV) and detected by QE-plus using nano-spray ion source (NSI). For full MS scan, peptides with 350–1700 *m/z* were detected at the resolution 70000. The automatic gain control (AGC) target was set to $1 × 10^6$, and maximum ion injection time was set to

50 ms. The ions with intensity above $2 \times 10^4$ were subjected to fragmentation via high energy collision induced dissociation (HCD) with 27% normalized collision energy (NCE). The fragmented ions were analyzed in Orbitrap. The AGC target in the ion trap was set to $1 \times 10^5$, and the isolation window was 1.6 *m/z*.

All data analysis was conducted using Thermo Scientific Proteome Discoverer 2.2. The *bovine* CSN2 or *human* PRSS35 in uniprot database was downloaded as target protein database. Enzyme specificity with trypsin was used, and maximum missing cleavages was set as two. The mass error tolerance for precursor ions was 20 ppm and fragment ions was 0.5 Da. False discovery rate (FDR) thresholds for protein and peptide were set at 0.01. For bovine CSN2, The fixed modification was set as Carbamidomethyl (C) and variable modifications were oxidation (M), acetylation (Protein N-term, K, S, T, Y), dimethylation (K), phosphorylation (K, S, T, Y). For *human* PRSS35, The fixed modification was set as Carbamidomethyl (C) and variable modifications were oxidation (M), acetylation (Protein N-term).

### Growth curves for liver cancer cell lines
Cells at different passages were seeded into 12-well plates at a density of 20,000 cells for different cells. Cell numbers in different wells were counted at Day 1 (24 h after seeding cells), 2, 3, 4, 5.

### Immunohistochemistry
Tissues were fixed in neutral buffered formalin for 24 h at room temperature and then embedded and processed according to standard protocols[81,82]. In brief, samples were dewaxed with xylene and then rehydrated with graded ethanol. After antigen retrieval, sections were incubated with 0.3% hydrogen peroxide for 10 min to block endogenous peroxidase activity. Next, sections were preincubated in normal goat serum for 15 min to prevent nonspecific staining, then samples were incubated with human C-PRSS35 or mouse Ly6G antibodies at room temperature. Four hours later, secondary antibody was used followed by incubation with DAB Chromogen dilution solution. IHC staining was quantitatively analyzed with the AxioVision Rel.4.6 computerized image analysis system assisted with the automatic measurement program (Carl Zeiss, Oberkochen, Germany). Ten representative staining fields of each section were analyzed to verify the Mean Optical Density (MOD). The MOD data were statistically analyzed by t-test to compare the average MOD difference between different groups of tissues. In addition, in IHC staining by anti-Ly6G antibody, the staining area/total area indicated the level of neutrophils infiltration.

### RNA sequencing and data process
Total RNA was extracted using Trizol (Life Technologies) according to the manufacturer's instructions. RNA integrity was assessed by RNA integrity number and determined using an Agilent 2100 Bioanalyzer. A total amount of 3 μg of RNA per sample was used for analysis. Sequencing sampling was performed from one single replicate. Libraries were generated using a NEBNext Ultra RNA Library Prep Kit for Illumina (NEB). RNA-seq was performed on an Illumina-HiSeq2500 platform by Novogene (Tianjin). Reads were aligned to the human genome hg19 (https://www.ncbi.nlm.nih.gov/assembly/GCF_000001405.13/). TopHat2 v.2.1.0, cufflinks v.2.2.1 and Integrative Genomics Viewer (IGV 2.10.3) were used to analyze RNA-seq data.

### Label-free secretome analysis
Culture cells to harvest at least 100 μg of secreted proteins per condition. Determine the protein concentration of the supernatant using established methods such as the Bradford Protein Assay Kit (Sangon biology). Guanidine hydrochloride were added to the supernatant to a final concentration of 4 M for secreted proteins denature. Then the proteins were reduced with 5 mM dithiothreitol (DTT) at 37 °C for 45 min and alkylated with 15 mM iodoacetamide (IAA) in the dark at

room temperature for 30 min. The alkylation was quenched by DTT at the final concentration of 20 mM at room temperature. Then 100 μg protein mixture were transferred to the 3kD molecular weight cut-off filter, spun down and washed twice with 100 mM HEPES buffer (pH 8.0). Then the proteins were resuspended in 100 μl 50 mM HEPES (pH 8.0) in the filter and digested by trypsin at a protein to enzyme ratio of 50:1 (w/w) (overnight, 37 °C). Then the digested peptides were filtered with 10kD molecular weight cut-off filter, desalted using Pierce C18 Spin Columns (Thermo Fisher Scientific). Peptide samples were vacuum dried for HPLC–MS/MS analysis.

For the HPLC–MS/MS analysis, the peptides were analyzed on an QE-plus mass spectrometer (Thermo Fisher Scientific, Waltham, MA, USA) equipped with an EASY-nLC 1200 nano liquid chromatography (HPLC) system (Thermo Fisher Scientific, Waltham, MA, USA). The peptides were dissolved in mobile phase buffer A (0.1% formic acid (FA) in water) and loaded onto Precolumn (Nano Trap 100 μm i.d. × 20 mm, Acclaim PepMap100 C18, 5 μm, 100 Å) and Analytical column (PepMap C18 2 μm 50 μm × 150 mm NV FS 1200 bar) by an autosampler. Then the peptides were separated and eluted with a linear gradient of 3–8% of mobile phase buffer B (0.1% FA in 80% acetonitrile) for 3 min, 8–28% buffer B for 70 min, 28–45% buffer B for 35 min, 45–90% buffer B for 2 min, and 90% buffer B for 10 min at the flow rate of 300nL/min on HPLC system. The eluted peptides were ionized under high voltage (2.2 KV) and detected by QE-plus using nano-spray ion source (NSI). For full MS scan, peptides with 350–1700 *m/z* were detected at the resolution 70000. The automatic gain control (AGC) target was set to $1 \times 10^6$, and maximum ion injection time was set to 50 ms. The ions with intensity above $2 \times 10^4$ were subjected to fragmentation via high energy collision induced dissociation (HCD) with 27% normalized collision energy (NCE). The fragmented ions were analyzed in Orbitrap. The AGC target in the ion trap was set to $1 \times 10^5$, and the isolation window was 1.6 *m/z*. The scan range fixed first mass at 110.0 *m/z*. The microscan was 1. The dynamic exclusion was set as 20 s. The charge exclusion were unassigned and charge states with 1, 7, 8, and >8.

All data analysis was conducted using MaxQuant (version 1.5.8.5) with Andromeda search engine. The reviewed human proteins documented with key word "secreted" in Uniprot database was downloaded as target human secreted protein database, the reviewed human proteins in Uniprot database was downloaded as target human protein database. First search peptide mass tolerance was 20 p.p.m. and main search peptide mass tolerance was 4.5p.p.m. MS/MS match tolerance was set to 0.5 Da. For the peptide identification via peptide spectrum matching the FDR was controlled with a standard target-decoy approach. 1% peptide FDR was applied at PSM level. Enzyme specificity with trypsin was used, and maximum missing cleavages was set as two. The fixed modification was set as Carbamidomethyl (C) and variable modifications were oxidation (M), acetylation (Protein N-term). False discovery rate (FDR) thresholds for protein and peptide were set at 0.01.

### SILAC-labeled secretome profiling
Collected secreted proteins and quantified using Bradford Protein Assay Kit (Sangon biology). For quantitative analysis, 200 μg proteins of both samples were combined prior to subsequent procedures. Guanidine hydrochloride were added to the lysate to a final concentration of 4 M for secreted proteins denature. Then the proteins were reduced with 5 mM dithiothreitol (DTT) at 37 °C for 45 min and alkylated with 15 mM iodoacetamide (IAA) in the dark at room temperature for 30 min. The alkylation was quenched by DTT at the final concentration of 20 mM at room temperature. Then 100 μg protein mixture were transferred to the 3kD molecular weight cut-off filter, spun down and washed twice with 100 mM HEPES buffer (pH 8.0). Then the proteins were resuspended in 100 μl 50 mM HEPES (pH 8.0) in the filter and digested by trypsin at a protein to enzyme ratio of 50:1

(w/w) (overnight, 37 °C). Then the digested peptides were filtered with 10kD molecular weight cut-off filter, desalted, and fractionated into four fractions using Pierce High pH Reversed-Phase Peptide Fractionation Kit (Thermo Fisher Scientific). Peptide samples were vacuum dried for HPLC–MS/MS analysis.

For the HPLC-MS/MS analysis, the peptides were analyzed on an QE-plus mass spectrometer (Thermo Fisher Scientific, Waltham, MA, USA) equipped with an EASY-nLC 1200 nano liquid chromatography (HPLC) system (Thermo Fisher Scientific, Waltham, MA, USA). The peptides were dissolved in mobile phase buffer A (0.1% formic acid (FA) in water) and loaded onto Precolumn (Nano Trap 100 μm i.d. × 20 mm, Acclaim PepMap100 C18, 5 μm, 100 Å) and Analytical column (PepMap C18 2 μm 50 μm × 150 mm NV FS 1200 bar) by an auto-sampler. Then the peptides were separated and eluted with a linear gradient of 3%–8% of mobile phase buffer B (0.1% FA in 80% acetonitrile) for 3 min, 8–28% buffer B for 70 min, 28–45% buffer B for 35 min, 45–90% buffer B for 2 min, and 90% buffer B for 10 min at the flow rate of 300nL/min on HPLC system. The eluted peptides were ionized under high voltage (2.2 KV) and detected by QE-plus using nano-spray ion source (NSI). For full MS scan, peptides with 350–1700 $m/z$ were detected at the resolution 70,000. The automatic gain control (AGC) target was set to $3 \times 10^6$, and maximum ion injection time was set to 50 ms. The ions with intensity above $2 \times 10^4$ were subjected to fragmentation via high energy collision induced dissociation (HCD) with 27% normalized collision energy (NCE). The fragmented ions were analyzed in Orbitrap. The AGC target in the ion trap was set to $1 \times 10^5$, and the isolation window was 1.6 $m/z$. The scan range fixed first mass at 110.0 $m/z$. The microscan was 1. The dynamic exclusion was set as 20 s. The charge exclusion were unassigned and charge states with 1, 7, 8, and >8.

For the SILAC-labeled proteome profiling, the raw data were searched with Thermo Scientific Proteome Discoverer 2.2 with Seaquest HT (v1.17) search engine. The proteins documented with key word "secreted" in uniprot database was downloaded as target Human secreted protein database. SILAC 2plex was set as quantification method. Enzyme specificity with trypsin was used, and maximum missing cleavages was set as two. The mass error tolerance for precursor ions was 20 ppm and fragment ions was 0.5 Da. The fixed modification was set as Carbamidomethyl (C) and variable modifications were oxidation (M), acetylation (Protein N-term). False discovery rate (FDR) thresholds for protein and peptide were set at 0.01.

## High-throughput protease screen

HTPS was performed as previously described[38]. In brief, we harvested PLC, HepG2, Hep3B and 293T cells, respectively. For lysis to achieve native lysate, we used HNN buffer (50 mM HEPES, 150 mM NaCl, 50 mM NaF, pH 7.8) supplemented with 0.5% NP-40 and protease inhibitor cocktail. Afterwards, the lysate was centrifuged at 14,000 × g for 15 min to remove any non-soluble material and the buffer was exchanged for 20 mM ammonium bicarbonate pH 7.8 using a filter device with molecular weight cut-off of 3kD.

Afterward, native cell lysate standardized in 20 mM Ammonium bicarbonate pH 7.8 was added at a final 50 μg of total protein per tube and mixed with the purified PRSS35 or PRSS35-domain1 at 1/50 [E]/[S] ratio. The samples were incubated at 37 °C for 12 h and collected by a 15 min centrifugation at 1200 × g in a low-binding tube. The collection step was repeated by adding 100 μl of MS-grade water. The fractions were transferred to low-binding tubes (Eppendorf) and concentrated on the speed-vacuum to complete dryness. The samples were stored at −80 °C until analysis. Before analysis, the samples were resuspended in 20 μl of MS-grade water with 0.1% formic acid and the peptide concentration was determined with Nanodrop UV spectrometer. The sample concentration was adjusted to 1 μg/μl with water containing 0.1% formic acid.

For the HPLC–MS/MS analysis, the peptides were analyzed on an QE-plus mass spectrometer (Thermo Fisher Scientific, Waltham, MA, USA) equipped with an EASY-nLC 1200 nano liquid chromatography (HPLC) system (Thermo Fisher Scientific, Waltham, MA, USA). The peptides were dissolved in mobile phase buffer A (0.1% formic acid (FA) in water) and loaded onto Precolumn (Nano Trap 100 μm i.d. × 20 mm, Acclaim PepMap100 C18, 5 μm, 100 Å) and Analytical column (PepMap C18 2 μm 50 μm × 150 mm NV FS 1200 bar) by an auto-sampler. Then the peptides were separated and eluted with a linear gradient of 3–8% of mobile phase buffer B (0.1% FA in 80% acetonitrile) for 3 min, 8–28% buffer B for 70 min, 28–45% buffer B for 35 min, 45–90% buffer B for 2 min, and 90% buffer B for 10 min at the flow rate of 300nL/min on HPLC system. The eluted peptides were ionized under high voltage (2.2 KV) and detected by QE-plus using nano-spray ion source (NSI). For full MS scan, peptides with 350–1700 $m/z$ were detected at the resolution 70,000. The automatic gain control (AGC) target was set to $1 \times 10^6$, and maximum ion injection time was set to 50 ms. The ions with intensity above $2 \times 10^4$ were subjected to fragmentation via high energy collision induced dissociation (HCD) with 27% normalized collision energy (NCE). The fragmented ions were analyzed in Orbitrap. The AGC target in the ion trap was set to $1 \times 10^5$, and the isolation window was 1.6 $m/z$. The scan range fixed first mass at 110.0 $m/z$. The microscan was 1. The dynamic exclusion was set as 20 s. The charge exclusion were unassigned and charge states with 1, 7, 8, and >8.

The raw data was searched with MaxQuant (version 1.5.8.5) with Andromeda search engine using the reviewed human UniProt database. We set the digestion mode to unspecific and the maximal peptide length to 40AA as described elsewhere[37,38]. Acetylation (N-termini) and oxidation (M) were set as variable modifications. First search peptide mass tolerance was 20 p.p.m. and main search peptide mass tolerance was 4.5 p.p.m. MS/MS match tolerance was set to 0.5 Da. For the peptide identification via peptide spectrum matching the FDR was controlled with a standard target-decoy approach. 1% peptide FDR was applied at PSM level and only peptide hits with a PEP score ≤0.05 and a score >40 were retained for further analysis. The final list of proteins was the union of proteins identified in the respective samples and we included only proteins with a global protein PEP ≤ 0.01 into the final database. Potential contaminants were excluded from the subsequent data analysis.

The protease cleavage window was used for the protease specificity determination using the iceLogo tool[39]. The reference set used for the calculation of the chance of the amino acid occurrence ($P$ value) at a certain position was the human reference proteome.

## Mice

Mice were housed at the animal facility of University of Science and Technology of China. Four-week-old male C57BL/6J and ICR mice were purchased from SLAC ANIMAL COMPANY. Four-week-old male Balb/c-nude mice were purchased from Beijing Vital River Laboratory Animal Technology Co., Ltd. *PRSS35*−/− mice (C57BL/6J) were generated using CRISPR genome editing (target sequence: 5′-AACGAGGGTACCGCTGCAGC-3′ and 5′-ACTCGGAACAGCAGCGTAAA-3′) and were obtained from the animal facility of the University of Science and Technology of China.

## Animal studies

All animals were housed at a suitable temperature (22–24 °C) and humidity (40–70%) under a 12/12-h light/dark cycle with unrestricted access to food and water for the duration of the experiment. All animal studies were conducted with approval from the Animal Research Ethics Committee of the University of Science and Technology of China. For syngeneic mouse tumor models, each group of Hepa1-6 cells ($4 \times 10^6$) was injected subcutaneously into four-week-old male mice (C57BL/6J; SLAC ANIMAL COMPANY). Tumor volumes were calculated using the

following formula: width (mm) × depth (mm) × length (mm) × 0.52. For xenograft model, each group of HepG2 cells ($4 × 10^6$) was injected subcutaneously into four-week-old male nude mice (BALB/c-nude mice; Beijing Vital River Laboratory Animal Technology Co., Ltd). The tumor burden was less than the maximum diameter (15 mm) approved by the Animal Research Ethics Committee of the University of Science and Technology of China. For YAP-5SA-induced-HCC models, plasmid DNA suspended in sterile Ringer's solution in a volume equal to 10% of the body weight was injected in 5–7 s via the tail vein of four-week-old male C57BL/6J or ICR (SLAC ANIMAL COMPANY) mice. The amount of injected DNA was 50 µg of per transposon plasmids together with 10 µg of PB transposase plasmids. At the end of animal studies, all mice were euthanized by inhaling carbon dioxide.

### Flow cytometry

For mouse livers and tumors, samples were cut into pieces and treated with collagenase IV and DNase I for one hour. The cell suspension was passed through a 70 µm cell strainer (Corning). The single-cell suspension was centrifuged at $300 × g$ for 5 min at 4 °C. The cell pellet was then resuspended in FACS buffer (1% BSA in PBS) and blocked with CD16/32 for 20 min, followed by incubating with the indicated antibodies for 30 min on ice. The signal was detected by using a BD Fortessa with BD FACSDiva Software (V8.03) and was analyzed using Flowjo v10 software. FACS sequential gating strategies for FACS experiments in Fig. 4c, Supplementary Fig 3j, m, Supplementary Fig 4c, h were presented in Supplementary Fig 4o.

### Clinical human tissue specimen

The 7 paired HCC lesions and the adjacent noncancerous clinical tissue samples were collected from HCC patients. The normal blood samples were collected from heathy physical examination persons. And the blood samples with HCC were collected from HCC patients. All the patients and physical examination persons are from the general surgery department, The first affiliated hospital of University of Science and Technology of China. The Supplementary Table 2 documented the clinicopathological characteristics of the 158 HCC patients used for the IHC assay (Supplementary Fig 1e). And the Supplementary Table 5 documented the basic characteristics of the HCC patients and normal subjects whose sera were used for the ELISA assay (Fig. 1f). We used two different batches of clinical patient specimens for the IHC analysis and the ELISA assay. To use these clinical materials for research purposes, prior patients' written informed consents and approval from the Institutional Research Ethics Committee of the first affiliated hospital of University of Science and Technology of China were obtained.

### Detection of serum MPO–DNA

We detected serum MPO–DNA using a previously described capture ELISA method with slight modifications. 96-well microtiter plates were coated with 5 µg/ml anti-MPO monoclonal antibody (Proteintech, 22225-1-AP) as the capturing antibody overnight at 4 °C. After blocking in 1% BSA, patient serum together with peroxidase-labeled anti-DNA monoclonal antibody was added (component No.2 of the Cell Death Detection ELISA kit, Roche, 11774425001), incubated at room temperature for 2 h and then washed with PBS three times. The peroxidase substrate (Roche, 11774425001) was added. After incubation at 37 °C for 40 min, the optical density was measured at 405 nm using a microplate reader.

### Neutrophil isolation

To isolate neutrophils from peripheral blood of mice, whole blood was collected via cardiac puncture (1 ml per animal) and suspended in HBSS (2 ml per animal) with 15 mM EDTA. After centrifugation ($400 × g$, 10 min, 4 °C), white cells were resuspended in 2 ml HBSS with 2 mM EDTA. Then, the cells were centrifuged ($1500 × g$, 30 min, room temperature) in a three-layer Percoll gradient (78%, 69%, and 52%) without braking. Neutrophils enriched in the interface of 69% and 78% layers were confirmed to be of > 95% purity by flow cytometric analysis (Supplementary Fig. 3g).

### Neutrophil migration assays

$5 ×10^5$ freshly isolated mice neutrophils in RPMI 1640 were added to the upper chamber of transwell device (Corning, 3402), and a 1:1 mixture of RPMI 1640 and indicated cancer cell conditional media was added to the lower chamber as the chemoattractant. The migrated neutrophils in the lower chamber were counted after three hours.

### PRSS35 in vitro substrate assay

To test for the cleavage of β-casein, PRSS35, PRSS35-domain1, PRSS35-domain2, or PRSS35-domain3 (0.025 mM final concentration) and β-casein (25 µM final concentration) were both added to a tube, respectively. Reactions were incubated for 12 h at 37 °C followed detection by WB.

To test for the cleavage of the substrates Dabcyl-KKKK-Edans, Dabcyl-LKKE-Edans and Dabcyl-RINKKIEK-Edans, a final concentration of 25 µM of each substrate was used. PRSS35-domain1 (0.025 mM final concentration) was added to a 96-well black, clear-bottom plate in 99 µl PBS. Substrates were dissolved in DMSO at 100× the indicated concentration (25 µM final concentration) and 1 µl was added to each well. Reactions were incubated for 12 h at 37 °C while being read on a CLARIOstar plate reader per hour (Molecular Devices) using an excitation wavelength of 340 nm and an emission wavelength of 490 nm.

To test for the cleavage of the substrate CXCL2, a final concentration of 25 µM of purified *human* CXCL2 was used. PRSS35-domain1 (0.5 mM final concentration) and *human* CXCL2 were both added to a tube. Reactions were incubated for 48 h at 37 °C followed detection by WB.

### Statistics and reproducibility

The data are presented as either mean ± s.d. or mean ± s.e.m. as stated. Statistical analyses were performed using either Prism8 (GraphPad Software) or SPSS software Statistics 22 (IBM Corp.). Data distribution was assumed to be normal, but this was not formally tested. Two-tailed unpaired Student's $t$-test and one-, two-analysis of variance (ANOVA) were used to calculate $P$-values. The Tukey method was used to adjust multiple comparisons. Kaplan–Meier curves were used to depict survival function from lifetime data for human patients using the log-rank test. The relationship between expression of PRSS35 and clinicopathological characteristics was analyzed by chi-square test. $P < 0.05$ was considered significant. The experiments were not randomized, except that mice were randomly grouped before different treatments. Data collection and analysis were not performed blind to the conditions of the experiments, except for IHC score analysis. Each experiment was repeated at least three times independently.

### Reporting summary

Further information on research design is available in the Nature Portfolio Reporting Summary linked to this article.

## Data availability

RNA-seq data generated in this study have been deposited in the Gene Expression Omnibus under accession code GSE201311. The mass spectrometry proteomics data generated in this study have been deposited to the ProteomeXchange Consortium via the PRIDE[83] partner repository with the dataset identifier PXD033437 and PXD033579. All data in the article, supplementary Information are available. Source data are provided with this paper.

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

## Acknowledgements

This work is supported in part by National Natural Science Foundation of China (81930083, 82192893, 91957203, 82130087, 81821001, 82341013), the Chinese Academy of Sciences (XDB39000000), National Key R&D Program of China (2022YFA1304504, 2018YFA0800300), Hefei Comprehensive National Science Center Institute of Health and Medicine Project (DJK-LX-2022001), the Fundamental Research Funds for the Central Universities (YD2070002008). Please address all the correspondence and requests for materials to H.Z. (hzhang22@ustc.edu.cn).

## Author contributions

H.Z. and P.G. conceived the study and supervised experiments. T.W., Y.Z, P.G. and H.Z. designed experiments. T.W., Y.Z, Z.Z, R.Y., P.Z., W.M., S.S., H.Y.L., J.F., H.L., L.Y., Y.C. and G.W. performed experiments. S.S. analyzed RNA-seq data. W.J. and Z.C. provided clinical specimens. L.S., T.Z. and X.Z. provided constructive advice. H.Z., P.G., T.W. and Y.Z. wrote the paper. All authors read and approved the manuscript.

## Competing interests

The authors declare no competing interests.
