## [Peer Review File · Nature Communications]

Secreted protease PRSS35 suppresses hepatocellular carcinoma by disabling CXCL2-mediated neutrophil extracellular trapsREVIEWER COMMENTS

Reviewer #1 (Remarks to the Author): with expertise in cancer, neutrophils, NETs

In this manuscript Wang and colleagues identify a secreted protease (PRSS35) which expression is lost in HCC cancer cells as compared to hepatocytes. Then the authors show that the levels of the protease are significantly decreased in patient sera as compared to healthy donors and also in HCC tissue as compared to matched healthy livers. They perform additional experiments that try to define the proteolytic events needed for the protease activation and run studies on potential cleavable targets for this protease. Finally, the authors identify CXCL2 as a potential target of the PRSS35 protease. In both subcutaneous and hydrodynamic injection-based HCC models they show that PRSS35 expression impairs tumor growth even in the presence of CXCL2 overexpression suggesting that loss of the PRSS35 protease is an important event in HCC tumor progression. Mechanistically, PRSS35 overexpression leads to less neutrophil infiltration and less signs of NETosis in HCC tumors. This is a very interesting study with important clinical implications. It highlights an important fact that is usually neglected in cancer, the fact that tumor transformation can lead to the downregulation of important homeostatic cytokines that can help tumor progression. The clinical data presented seems robust and again highlight the importance of CXCR1/2 agonist chemokines in tumor progression and their relevance as potential targets in cancer, despite the little success that inhibitors against this route have had in the clinic. The authors have performed extensive experimentation to test their hypothesis and they have even generated a new KO model. Yet, some points raised my need additional evidence and some experiments may be needed to really prove the translational implications of their findings.

Major points.

-Other groups have recently shown the implications of the CXCR1/2 axis in the development of NAFLD and subsequent HCC (Lesley, Mackey, Gut 2022 and others). As NAFLD is a major cause for HCC (although the authors do not see evident correlation with cirrhosis) the authors should try to look for some kind of evidence for the loss of PRSS35 in progressive NAFLD. For example, do the protein decrease in the sera of mouse models of NAFLD or in patients with fatty liver, or can the authors identify patients with such aetiology among their HCC patients and look for potential correlations. I understand that this may represent a new line of work for the group and experiments or sample identification and obtention may result difficult. Yet, I think that there are great chances to find that fatty liver hepatocytes may lose PRSS35 expression and that the mechanism identified by the authors may play a role in such condition. The authors should at least intensively discuss about these issues and provide additional citations as the one suggested.

- the authors develop a new ELISA method to identify circulating active PRSS35, due to the importance of the data in HCC patients the authors should provide in the supplementary information some sort of validation of the in house developed ELISA (linearity of the standard curves, sensitivity,...).

- The authors develop a new mouse model that is a KO for PRSS35 to test if in such models hydrodynamic injection drives faster development of HCC. These are very interesting experiments but no background information is provided in the phenotype of these mice, are these mice healthy, do they have basal liver damage (aminotransferases, basal Ly6G infiltration). Some info on the phenotype of the mice is required for proper interpretation of the data.

-Another important point that remains unanswered in the paper is if hPRSS35 targets and degrades also members of the human CXCL1 chemokine family. In humans CXCL2 may not be the most relevant member of the family supporting tumor progression. The authors should test degradation of human chemokines by hPRSS35. Do human chemokines conserve the KK targeted motif?

- Mice experiments indicate that mPRSS35 expression induces less CXCL2 and consequently less Ly6G infiltrate and less NETosis. The studies in Fig 4 are highly suggestive of decreased NETosis but at this point Cit H3 may not be enough evidence of NETosis (although is clearly diminished when neutrophils are depleted). This reviewer would recommend some sort of immunostaining

showing at least H3Cit colocalization with DNA and ideally include Ly6G to identify neutrophils or stain for neutrophil elastase with good antibodies working in mouse tissue. MPO/DNA in mouse sera is an ELISA that have some important issues regarding background and off-target detection. The authors should perform also a H3Cit commercial ELISA to confirm their data in the sera of the mice included in figure 4. Ly6G immunostaining of the tissue is not very convincing, may be is due to the quality and magnification of the images, the background is high, and the staining does not seem to identify individual membranes. At least some zoomed images should show the shape of the nuclei and the appropriate specificity of the staining. In addition to these technical issues, the absence of NETs could be just a direct result of less neutrophils and may not be directly responsible for tumor progression. Although is very difficult to design an experiment showing that the slower growth of tumors promoted by PRSS35 expression is due to NETosis the authors could try to replicate the experiments performed in panels H to K but treating the mice with DNase or PAD4 inhibitors and see if the effect on tumor growth of PRSS35 overexpression is similar to the effect of NET elimination.

Minor

In Fig 2, indicate if cleaved forms of PRSS35 in the coomassie gel correspond to the expected sizes according to LC-MS.

Reviewer #2 (Remarks to the Author): with expertise in cancer, pro-protein convertases

The manuscript from Wang T. and collaborators is focused on the expression and role of a PRSS35 protein involve in hepatocellular carcinoma (HCC) growth regulation through depletion of CXCL2 attenuating neutrophil recruitment to tumors and formation of neutrophil extracellular traps suppressing HCC progression. Activation of mature PRSS35 is under the control of a pro-protein convertase. The manuscript needs be re-written especially the legends of the figures. The manuscript lacks clarity and the figures needs to be edited, most of them have no labels and the legends are not complete and lack explanation. Furthermore, the manuscript present major issues and needs a major revision.

1- Statistical Rational for mass spectrometry analyses needs to be part of the manuscript. How many replicates have been performed? Heatmap with all the triplicate needs to be presented as well as all raw data need be put in Pride and excel data sheet in supp data.

2- For mass spectrometry shot gun analyses. What was the Default charge state set at 2, the unassigned and charge states, the dynamic exclusion, the scan range. For ddMS², what was the scan range, the microscan and the isolation window used?

3- For Mass spectrometry analyses what is the search engine, and the databank used. For the parameters what was the number of peptides per proteins, number of miss cleavage. What was the mass tolerance for the precursors, and the fragments? For the relative label free quantification. What were the parameters set for peptide per protein, miss cleavage the FDR and again the mass tolerance?

4- Figure 1 : where are a, c, d, e labels... Please, explain what is EV, Lys, Sup. For the mS/MS spectrum. What is the inset? The authors must provide in supp data other MS/MS for the proteins identified.

5- For the pro-protein convertase (PC), it is necessary to identify the ones implicated. Furin is ubiquitous but some of PCs like PACE4 are well established in some cancer. The used of specific inhibitors or siRNA to identify the nature of PC are necessary to discriminate the specific PC involve in the HCC. Moreover, PC have a specific signature for their catalytic sites. It is thus necessary to identify such a signature for the PC involve in HCC.

6- For the CXCL2 cleavage at the di-basic residue as a trypsin like protein. It is necessary to demonstrate that PRSS35 is acting as a trypsin and the use of selective inhibitors and perform mutagenesis at the catalytic site level.

Reviewer #3 (Remarks to the Author): with expertise in HCC, neutrophils, NETs

In this manuscript by Wang et al, the authors employed a proteomic analysis of the secretome of HCC cells and identified a close relationship between the secreted protein PRSS35, NETs and HCC prognosis. Their findings suggest a potential value for PRSS35 as a biomarker for screening in early stage HCC. The main mechanism identified include HCC tumors cell have decreased secreted PRSS35 levels that lead to an increase in the chemokine CXCL2. CXCL2 results in greater neutrophil accumulation and increased NETs formation in the TME which is pro-tumorigenic. Overall, the authors have performed a considerable body of work, including the analysis of HCC patients, to demonstrate a novel and potentially biomarker role for the secreted protein PRSS35. The manuscript is well written and deeply investigates the mechanisms through characterizing the unique features of PRSS35 suppressing HCC progression by inhibiting CXCL2-mediated neutrophil recruitment and NETs formation.

Points to address:

1. The main conclusion is that PRSS35 suppresses HCC progression by inhibiting CXCL2-mediated neutrophil recruitment and NETs formation. However, current single-cell technologies show that neutrophils probably have a dynamic spectrum of pro- and anti-tumor (mostly N1 or High-density neutrophils) functions that vary according to their microenvironment (Nat Rev Gastroenterol Hepatol. 2022 Apr;19(4):257-273.). CXCR2+ is a maturation and activation marker of N1 neutrophils which was thought to play an anti-tumor role (Cell Rep. 2015 Feb 3;10(4):562-73.; Front Immunol. 2019 Aug 14;10:1912.; Nat Rev Cancer. 2020 Sep;20(9):485-503.). In this regard, the authors should supplement further analysis of which subpopulation of neutrophil was involved in this protective effect of PRSS35. The anti-tumor role of neutrophils should also be added in the discussion.
2. Have the authors overexpressed PRSS35 in human HCC cells lines in addition to murine Hepa1-6 cells? Confirming role of PRSS35 in vivo with a xenograft HCC model may be important in addition to the Hepa1-6-murine HCC model.
3. What was the rationale for using YAP-5SA tail vein injection induced HCC model? Have the authors confirmed the recruitment of neutrophils and NET formation in control and PRSS35 KO mice undergoing YAP-5SA HCC model in addition to just measuring number of tumors?
4. The finding that there is less secreted PRSS35 in malignant cells is novel and of great potential for future biomarker. What is the underlying mechanism/regulation of PRSS35 in normal/cancer/cancer cells?
5. The author mentioned that PRSS35 was the most significantly down-regulated protein in the PLC secretome among the 236 secreted proteins. Why did the authors focus on the down-regulated ones rather than up-regulated proteins?
6. The author mentioned that "they use ELISA to analyze the serum from 149 HCC patients and 73 normal subjects" and the numbers for the information of HCC patients as well as healthy donors in supplementary Table 4 are 149 and 73. However, the number of HCC patients for clinicopathological characteristics in supplementary table2 is 158. Please carefully clarify this mismatch.

Reviewer #4 (Remarks to the Author): with expertise in cancer, proteomics, secretome

This manuscript by Wang et al. reports the discovery of PRSS35 serine protease as a secreted protein that is downregulated in HCC and acts as a tumor suppressor through downregulation of the chemokine CXCL2. Overall, the authors provide convincing evidence that PRSS35 is involved in HCC through in vitro and in vivo experiments. I feel that this manuscript is suitable for publication after the following issues have been addressed:

Major issues

1. The authors used a single cell line representing normal hepatocytes. They should test other normal hepatocyte cell lines as well as primary hepatocytes as well to confirm their findings in a larger set of samples.

2. The authors do not provide coverage of the different versions of the PRSS35 protein that they obtained by MS/MS based peptide sequencing. They just show a 'representative' peptide in Figure 1d. The size difference could also potentially be due to glycosylation as a major post-translational modification of secreted proteins or because of the epitope for antibody-based detection being altered by glycosylation.

3. Overall, there is excessive reliance on western blotting to elucidate the nature of the different forms - I suggest that the author use LC-MS/MS analysis to characterize all isoforms that they detected by western blotting. This includes the forms that they see in serum (Figure 1e) as well as in cell line experiments (Figure 2e).

4. I have an issue with the database search strategy used by the authors. In the methods (lines 534-535), it seems that they used only a subset of the human protein database. This is wrong for two reasons: i) many proteins that are not secreted can be found in the secretome because of a small number of cells dying; ii) many proteins that are membrane bound can be found in the secretome; and, iii) many proteins that are secreted as not properly annotated as 'secreted.' The authors should search against the entire human protein database.

Point-by-point response to the comments

Response to all reviewers:

We thank the referees for their-constructive and insightful comments and suggestions that have helped us with the revision. In the past six months, we have performed a significant number of experiments in multiple *in vivo* and *in vitro* models, and have addressed all the concerns and comments raised by our editor and referees. Here we resubmit a substantially improved manuscript along with our point-by-point responses.

For the referee's convenience, we have appended in this file all the revised figures, which we labeled as **Figure R1** to **Figure R18**.

Reviewer Comments:

Reviewer #1 (Remarks to the Author):

In this manuscript Wang and colleagues identify a secreted protease (PRSS35) which expression is lost in HCC cancer cells as compared to hepatocytes. Then the authors show that the levels of the protease are significantly decreased in patient sera as compared to healthy donors and also in HCC tissue as compared to matched healthy livers. They perform additional experiments that try to define the proteolytic events needed for the protease activation and run studies on potential cleavable targets for this protease. Finally, the authors identify CXCL2 as a potential target of the PRSS35 protease. In both subcutaneous and hydrodynamic injection-based HCC models they show that PRSS35 expression impairs tumor growth even in the presence of CXCL2 overexpression suggesting that loss of the PRSS35 protease is an important event in HCC tumor progression. Mechanistically, PRSS35 overexpression leads to less neutrophil infiltration and less signs of NETosis in HCC tumors. This is a very interesting study with important clinical implications. It highlights an important fact that is usually neglected in cancer, the fact that tumor transformation can lead to the downregulation of important homeostatic cytokines that can help tumor progression. The clinical data presented seems robust and again highlight the importance of CXCR1/2 agonist chemokines in tumor progression and their relevance as potential targets in cancer, despite the little success that inhibitors against this route have had in the clinic. The authors have performed extensive experimentation to test their hypothesis and they have even generated a new KO model. Yet, some points raised my need additional evidence and some experiments may be need

to really prove the translational implications of their findings.

Response: We appreciate the referee's positive comments that well summarized the major findings and significance of our study. We thank the referee for the suggestions and have provided multiple additional data to strengthen our conclusions in the revised manuscript.

Major points.

Comments 1-1:

-Other groups have recently shown the implications of the CXCR1/2 axis in the development of NAFLD and subsequent HCC (Lesley, Mackey, Gut 2022 and others). As NAFLD is a major cause for HCC (although the authors do not see evident correlation with cirrhosis) the authors should try to look for some kind of evidence for the loss of PRSS35 in progressive NAFLD. For example, do the protein decrease in the sera of mouse models of NAFLD or in patients with fatty liver, or can the authors identify patients with such aetiology among their HCC patients and look for potential correlations. I understand that this may represent a new line of work for the group and experiments or sample identification and obtention may result difficult. Yet, I think that there are great chances to find that fatty liver hepatocytes may lose PRSS35 expression and that the mechanism identified by the authors may play a role in such condition. The authors should at least intensively discuss about these issues and provide additional citations as the one suggested.

Response 1-1: This is a very good point. As the reviewer has pointed out, CXCR1/2 axis is involved in NAFLD and mediates neutrophil function which contributed to the development of HCC in NAFLD (Yang, Luo et al. 2020, Wang, Zhang et al. 2021, Kotsiliti 2022, Leslie, Mackey et al. 2022), it is possible that PRSS35 could suppress the development of NAFLD and subsequent HCC through the CXCL2-neutrophil axis. Following the reviewer's suggestion, we determined the PRSS35 levels in serums from NAFLD patients compared to healthy subjects by WB and ELISA assay. Our results showed that PRSS35 was also decreased in NAFLD patients' serum compared to healthy subjects (**Figure R1A and 1B**). Therefore, these results indicated that not only HCC, but also NAFLD is associated with low PRSS35 levels, suggesting the potential role of PRSS35 in the development of NAFLD. Following the reviewer's suggestion, we have discussed this part and provided additional citations in the discussion section of the revised manuscript.

Figure R1. PRSS35 is downregulated in NAFLD. (A) The serum samples from normal subjects and NAFLD patients were collected. The PRSS35 levels in these serum samples were determined by WB with three PRSS35 antibodies. (B) Serum PRSS35 protein levels were measured by customized ELISA kit from 10 normal subjects and 10 human NAFLD patients. Data are presented as the mean \pm s.e.m.

Comments 1-2:

- the authors develop a new ELISA method to identify circulating active PRSS35, due to the importance of the data in HCC patients the authors should provide in the supplementary information some sort of validation of the in house developed ELISA (linearity of the standard curves, sensitivity,...).

Response 1-2: Following the reviewer's suggestion, we provided supplementary information for our in house developed PRSS35 ELISA kit, which included the linearity information (**Figure R2A**) and the accuracy information (**Figure R2B**). The sensitivity of this kit is < 0.5 ng/mL and the detection concentration range is 0.5 ng/mL – 50 ng/mL. We also included the linearity data as **Extended Figure 1h** in the revised manuscript.

Figure R2. Supplementary information of PRSS35 ELISA kit. (A) The standard curve of PRSS35 ELISA kit. We set ten diluted concentrations of purified PRSS35 domain1 protein fragment and determined the corresponding OD450 to calculate the linearity information. **(B)** The accuracy information of PRSS35 ELISA kit. We determined three samples with different concentrations for eight times to calculate the intra- and inter-accuracy information of the PRSS35 ELISA kit. Please see also **Extended Figure 1h** in the revised manuscript.

Comments 1-3:

- The authors develop a new mouse model that is a KO for PRSS35 to test if in such models hydrodynamic injection drives faster development of HCC. These are very

interesting experiments but no background information is provided in the phenotype of these mice. are these mice healthy, do they have basal liver damage (aminotrasnfereases, basal Ly6G infiltration). Some info on the phenotype of the mice is required for proper interpretation of the data.

Response 1-3: We thank the reviewer for this very important suggestion. As a matter of fact, we had determined the background information of PRSS35-KO mice and further compared to corresponding WT mice at eight weeks of age. Our data showed that PRSS35-KO mice exhibited the similar physical development and metabolic capability as those of the WT mice (**Figure R3A-C**). Moreover, the liver morphology and basal liver damage of the PRSS35-KO mice were also normal compared to WT mice (**Figure R3D-F**). In addition, no significant differences were observed in the activity and appearance between the PRSS35-KO and the WT mice. Therefore, our data indicated that the PRSS35-KO mice were healthy and had the similar physical condition as the WT mice. We have included these data as **Extended figures 3e-j** in the revised manuscript.

Figure R3 Comparison of the background information of PRSS35-KO mice and WT mice at eight weeks of age. (A) Body, liver and spleen weight of WT and PRSS35-KO mice. **(B)** Triglyceride (TG), total cholesterol (TCHO) and glucose (Glu) in serum of WT and PRSS35-KO mice. **(C)** Aspartic transaminase (AST) and alanine aminotransferase (ALT) concentration in serum of WT and PRSS35-KO mice. **(D)** Photograph of spleen and liver of WT and PRSS35-KO mice. **(E)** Hematoxylin staining (H&E) of liver from WT and PRSS35-KO mice. **(F)** The neutrophil infiltration in liver of WT and PRSS35-KO mice. Please see also **Extended figures 3e-j** in the revised manuscript.

Comments 1-4:

-Another important point that remains unanswered in the paper is if hPRSS35 targets and degrades also members of the human CXCL1 chemokine family. In humans CXCL2 may not be the most relevant member of the family supporting tumor progression. The authors should test degradation of human chemokines by hPRSS35. Do human chemokines conserve the KK targeted motif?

Response 1-4: We appreciate the reviewer for the important point. Following the reviewer's suggestion, we first performed the alignment analysis of human CXCL family members, including CXCL1, CXCL2, CXCL3, CXCL5 and CXCL8. Results showed that CXCL1, CXCL2 and CXCL3 were highly homologous and all have targeted KK motif. But CXCL5 and CXCL8 did not (**Figure R4A**). We then determined their protein levels in the cell lysate and culture medium in PLC cells with overexpressed EV or hPRSS35. The results showed that hPRSS35 degraded CXCL2 as we previously described, and it also degraded CXCL1 and CXCL3. But CXCL2 is the most significant member degraded by PRSS35. Meanwhile, CXCL5 and CXCL8 could not be degraded by PRSS35, which is consistent with our findings that they bear no targeted KK motif (**Figure R4B**). Taken together, KK motif is not conserved in all CXCL family members, and the KK motif bearing CXCL2 is the most significant member degraded by PRSS35.

Figure R4. CXCL2 is the most significant member degraded by PRSS35 among CXCL family members. (A) Alignment of CXCL1, CXCL2, CXCL3, CXCL5, and CXCL8 protein sequences. All the targeted KK motifs were marked by red frame. (B) The lysate and culture medium of PLC cells stably expressing EV or PRSS35 were collected. CXCL family member levels in lysate and culture medium were determined by WB.

Comments 1-5:

- Mice experiments indicate that mPRSS35 expression induces less CXCL2 and consequently less Ly6G infiltrate and less NETosis. The studies in Fig 4 are highly suggestive of decreased NETosis but at this point Cit H3 may not be enough evidence of NETosis (although is clearly diminished when neutrophils are depleted). This reviewer would recommend some sort of immunostaining showing at least H3Cit colocalization with DNA and ideally include Ly6G to identify neutrophils or stain for neutrophil elastase with good antibodies working in mouse tissue. MPO/DNA in mouse sera is an ELISA that have some important issues regarding background and off-target detection. The authors should perform also a H3Cit commercial ELISA to confirm their data in the sera of the mice included in figure 4. Ly6G immunostaining of the tissue is not very convincing, may be is due to the quality and magnification of the images, the background is high, and the staining does not seem to identify individual membranes. At least some zoomed images should show the shape of the nuclei and the appropriate specificity of the staining. In addition to these technical issues, the absence of NETs could be just a direct result of less neutrophils and may not be directly responsible for tumor progression. Although is very difficult to design an experiment showing that the slower growth of tumors promoted by PRSS35 expression is due to NETosis the authors could try to replicate the experiments performed in panels H to K but treating the mice with DNase or PAD4 inhibitors and see if the effect on tumor growth of PRSS35 overexpression is similar to the effect of NET elimination.

Response 1-5: Following the reviewer's suggestions, we have performed multiple additional experiments.

(1) We first performed immunofluorescence experiment using previous paraffin section samples from the mice experiments in **Figure 4a** in the original manuscript to confirm the occurrence of NETosis in the mouse tumors. Immunostaining results showed that H3Cit was co-localized with DNA, which indicated that NETosis indeed existed in mouse tumors (**Figure R5A, B**). Moreover, mCXCL2 overexpression enhanced the formation of NETs, while tumors expressing mPRSS35 showed reduced NETs formation with or without mCXCL2 co-expression (**Figure R5A**). On the other hand, immunostaining experiments using the mouse tumor samples in **Figure 4h** in the original manuscript also revealed that mPRSS35 expression led to reduced NETs formation, and anti-Ly6G

treatment groups exhibited the lowest NETs formation with or without mPRSS35 overexpression (**Figure R5B**). These data suggested that mPRSS35 suppressed NETs formation in mouse tumors, which is consistent with the WB and ELISA results in **Figure 4b, 4d, and 4j, 4k** in the original manuscript. Taken together, our data demonstrated that NETosis indeed happened in the mouse tumors and that mPRSS35 expression induced less CXCL2 and consequently less Ly6G infiltrate and less NETosis.

(2) We also performed mouse H3Cit ELISA assay using a commercial kit. Consistent with MPO-DNA ELISA results in **Figure 4d** in the original manuscript, we found that H3Cit levels were markedly increased in the sera of mice harboring tumors from mCXCL2 overexpressing Hepa1-6 cells, but decreased in mice harboring tumors from mPRSS35 overexpressing Hepa1-6 cells with or without mCXCL2 overexpression (**Figure R6A**). Similar assay was also performed using the mouse serum samples from the experiments in **Figure 4k** in the original manuscript. The results showed that mPRSS35 expression significantly reduced the mouse serum H3Cit levels, and anti-Ly6G treatment resulted in very low serum H3Cit levels in both EV and mPRSS35 overexpressing groups (**Figure R6B**), which were consistent with MPO-DNA ELISA results in **Figure 4k** in the original manuscript. Therefore, following the reviewer's suggestion, we confirmed the NETs levels in sera through two canonical methods. We have included these data as **Extended Figure 4f** in the revised manuscript.

(3) We improved our Ly6G IHC staining experiments. In our new data **Figure R7A and 7B**, we could clearly observe the individual neutrophil staining by Ly6G antibody (**Figure R7A and R7B**, left panel). The updated statistical results (**Figure R7A and R7B**, right panel) were consistent with previous data. We have also added these data into **Figure 4e** and **Extended Figure 4i** in the revised manuscript to replace the original ones.

(4) In addition, we also treated the mice with DNase I rather than anti-Ly6G antibody, and replicated the experiments performed in **Figure 4h to k** in the original manuscript. The results showed that, similar as mPRSS35 overexpression, NETs elimination by DNase I treatment significantly suppressed the tumor growth (**Figure R8A**). As expected, the infiltration of neutrophils was not affected by DNase I, but was suppressed by mPRSS35 (**Figure R8B and R8E**). Importantly, NETs were significantly decreased by both mPRSS35 overexpression and DNase I treatment in mouse serum as well as tumor tissue samples (**Figure R8C and 8D**). Combing with the experiments using anti-Ly6G in

Figure 4h to k in the original manuscript, our data further consolidate that PRSS35 inhibits HCC development by suppressing CXCL2-mediated neutrophil NETs formation. We have included these data as **Extended Figure 4j-n** in the revised manuscript.

Figure R5. Immunofluorescent staining of NETs in mice tumors. (A) Neutrophils and NETs were detected by immunofluorescent staining with Ly6G and H3Cit antibodies using the mouse tumor samples from **Figure 4a** in the original manuscript. **(B)** Neutrophils and NETs were detected by immunofluorescence staining with Ly6G and H3Cit antibodies using the mouse tumor samples from

Figure 4h in the original manuscript. Neutrophils express Ly6G (red), NETs and NETs forming neutrophils express citrullinated histone H3 (green). DAPI serves as a nuclear DNA counterstain (blue). Cyan fluorescence represents the colocalization of citrullinated histone H3 with DNA (green overlapped with blue). The red arrowheads point to the neutrophils without NETs formation, and the white arrows point to the NETs and NETs-forming neutrophils. The dashed lines in the top panel highlight the magnified areas. Higher-magnification images of the boxed areas (middle row) are shown along with single channel images (bottom row); scale bars, 100 μ m.

Figure R6. The H3Cit ELISA analysis of serum NETs. (A, B) Serum NETs levels were measured by mouse H3Cit ELISA kit using the sera from the mice experiment in **Figure 4d** (A) and **Figure 4k** (B) in the original manuscript. Data are presented as the mean \pm s.e.m (A, B). Please see also **Extended Figure 4f** in the revised manuscript.

Figure R7. Immunohistochemical staining of neutrophils in mice tumors. (A) Representative IHC images of Ly6G expression in tumors from mouse experiments in **Figure 4a** in the original manuscript (left panel). Statistical quantification of Ly6G-positive area/total area in IHC assay (right panel). **(B)** Representative IHC images of Ly6G expression in tumors from mouse experiments in **Figure 4h** in the original manuscript (left panel). Statistical quantification of Ly6G-positive area/total area in IHC assay (right panel). We have also included these data as **Figure 4e** and **Extended Figure 4i** in the revised manuscript to replace the original ones.

Figure R8. PRSS35 inhibits HCC development by suppressing CXCL2-mediated neutrophil NETs formation (A) Hepa1-6 cells stably expressing EV or Flag-mPRSS35 were injected subcutaneously into C57BL/6J mice, followed by daily intraperitoneal injection of DNase I five days later. Photograph showed tumors at the end of the experiment (left panel). Tumors were extracted and weighted at the end of the experiment (right panel). (B) The frequency of neutrophil in tumors was detected by Flow cytometry. (C) Western blot analysis of mCXCL2, Flag-mPRSS5 and H3Cit protein levels in tumors. β -actin served as a loading control. (D) Serum NETs levels were measured by H3Cit ELISA kit. (E) Representative IHC images (left panel) and quantification (right panel) of neutrophils in the tumors using Ly6G staining. Data are presented as the mean \pm s.d. (A, B, D, E). Please see also **Extended Figure 4j-n** in the revised manuscript.

Comments 1-6

Minor

In Fig 2, indicate if cleaved forms of PRSS35 in the coomassie gel correspond to the expected sizes according to LC-MS.

Response: We thank the reviewer's insightful comments and questions. We performed LC-MS to determine the sequence of the smallest cleaved forms of PRSS35 in the coomassie gel in **Figure 2a** in the original manuscript. The results showed that two peptides of PRSS35 were identified (**Figure R9A**, upper panel), and that the two peptides were located in the PRSS35-Domain1 region, which agreed with the expected sizes (**Figure R9A**, lower panel).

Figure R9. Identification of the shortest cleaved forms of PRSS35 in coomassie gel by mass spectrometry. (A) The shortest cleaved forms of PRSS35 from **Figure 2a** in the original manuscript were analyzed by mass spectrometry. The sequences of identified peptides (upper panel) and their locations in PRSS35 protein (lower panel) were shown.

Reviewer #2 (Remarks to the Author):

The manuscript from Wang T. and collaborators is focused on the expression and role of a PRSS35 protein involve in hepatocellular carcinoma (HCC) growth regulation through depletion of CXCL2 attenuating neutrophil recruitment to tumors and formation of neutrophil extracellular traps suppressing HCC progression. Activation of mature PRSS35 is under the control of a prop-protein convertase. The manuscript needs be re-written especially the legends of the figures. The manuscript lacks clarity and the figures needs to be edited, most of them have no labels and the legends are not complete and lack explanation. Furthermore, the manuscript present major issues and needs a major revision.

Response: We appreciate the reviewer for the critical and constructive comments and suggestions. Accordingly, we have revised the results section and figure legends, and provided the completed labels and legends of all figures. More importantly, we have performed multiple additional experiments to strengthen our conclusions in the revised manuscript. We thank the reviewer for helping us submit a substantially improved manuscript.

Comments 2-1:

1- Statistical Rational for mass spectrometry analyses needs to be part of the manuscript. How many replicates have been performed? Heatmap with all the triplicate needs to be presented as well as all raw data need be put in Pride and excel data sheet in supp data.

Response 2-1: Thank our reviewer for the comments. Actually, we performed the proteomics experiment in **Figure 1a** in the original manuscript for multiple times. Following the suggestion, we presented the triplicate results using the Heatmap plot (**Figure R10A**). All proteomics data including raw data and excel data sheet shown in the heatmap and volcano plots were submitted to the ProteomeXchange Consortium via the PRIDE partner repository with the dataset identifier PXD033437, and also presented in **supplementary Table 7** in the revised manuscript. The excel data we submitted include the detailed LC-MS condition for proteomics experiments, the list of the peptides and proteins we identified, as well as the detailed LC-MS information of the identified peptides.

Figure R10. The heatmap analysis of the triplicate proteomics data. (A) Heatmap analysis of the triplicate secretomics data in THLE3 and PLC cells shown in **Figure 1a** in the original manuscript. Red indicates relative high level, whereas blue indicates relative low level. The data was normalized with Z-score and performed cluster analysis with Euclidean distance.

Comments 2-2:

2- For mass spectrometry shot gun analyses. What was the Default charge state set at 2, the unassigned and charge states, the dynamic exclusion, the scan range. For ddMS², what was the scan range, the microscan and the isolation window used?

Response 2-2: We appreciate your insightful comments.

(1) For mass spectrometry shot gun analyses, default charge state setting at 2 meant that bivalent peptides were mainly monitored. The unassigned and charge states with 1, 7, 8,

and >8 were charge exclusion. The dynamic exclusion was set as 20 seconds. Furthermore, the scan range of MS was 350m/z to 1700 m/z.

(2) For ddMS², the scan range fixed first mass at 110.0 m/z, the microscan was 1, and the isolation window was 1.6m/z.

We have added these information into the Materials and Methods section of the revised manuscript.

Comments 2-3:

3- For Mass spectrometry analyses what is the search engine, and the databank used. For the parameters what was the number of peptides per proteins, number of miss cleavage. What was the mass tolerance for the precursors, and the fragments? For the relative label free quantification. What were the parameters set for peptide per protein, miss cleavage the FDR and again the mass tolerance?

Response 2-3: We appreciate our reviewer for the insightful comments.

(1) For Mass spectrometry analyses, we performed the analyses with Andromeda search engine. The reviewed human proteins documented with key word “secreted” in Uniprot database was downloaded as target human secreted protein databank.

(2) For the parameters, the number of peptides per protein and the number of miss cleavage were documented in the **supplementary Table 7** in the revised manuscript.

(3) The mass tolerance for the precursors was 20p.p.m. and main search peptide mass tolerance was 4.5p.p.m. MS/MS match tolerance was set to 0.5Da.

(4) For the relative label free quantification, the number of peptides per protein and the number of miss cleavage were documented in the **supplementary Table 7** in the revised manuscript. False discovery rate (FDR) thresholds for protein and peptide were set at 0.01. The mass tolerance for the precursors was 20p.p.m. and main search peptide mass tolerance was 4.5p.p.m. MS/MS match tolerance was set to 0.5Da.

All the information has been added in the Materials and Methods section of the revised manuscript.

Comments 2-4:

4- Figure 1 : where are a, c, d, e labels... Please, explain what is EV, Lys, Sup. For the mS/MS spectrum. What is the inset? The authors must provide in supp data other MS/MS for the proteins identified.

Response 2-4: Thank you for the comments. EV, Lys and Sup indicate protein samples from cells expressing control empty vector (EV), from cell lysate and from supernatant, respectively. We have provided this information in the revised figure legend. The inset of the MS/MS spectrum was the precursor's MS spectrum. The other MS/MS data for PRSS35 we identified have been provided in **supplementary Table 8** in the revised manuscript.

Comments 2-5:

5- For the pro-protein convertase (PC), it is necessary to identify the ones implicated. Furin is ubiquitous but some of PCs like PACE4 are well established in some cancer. The used of specific inhibitors or siRNA to identify the nature of PC are necessary to discriminate the specific PC involve in the HCC. Moreover, PC have a specific signature for their catalytic sites. It is thus necessary to identify such a signature for the PC involve in HCC.

Response 2-5: Following the reviewer's suggestion, we investigated the effect of the known pro-protein convertases (PCs) on PRSS35 by using their specific shRNAs. The results showed that, besides PCSK3 (also named as FURIN), knocking down PCSK1 also resulted in increased FL-PRSS35 and decreased SF-PRSS35, suggesting that both PCSK1 and PCSK3 could cleave PRSS35 (**Figure R11A and R11B**). Meanwhile, we also determined PCSK1 levels in HCC tumor tissues and their corresponding non-cancerous adjacent liver tissues. The results showed that PCSK1 was down-regulated in HCC tissues, which was consistent with reduced PRSS35 activity in HCC (**Figure R11C**). These data indicated that PCSK1 was possibly involved in HCC development by cleaving PRSS35. Moreover, as the reviewer mentioned, each PC has a specific signature for its catalytic sites (Remacle, Shiryaev et al. 2008, Tian, Huajun et al. 2012). To further uncover the signature of PCSK1 in HCC, we analyzed the new-generated peptide terminals from PCSK1-cleaved PRSS35 and used these terminal sequences to calculate the frequency of the individual residue occurrence at individual

substrate positions relative to the P1-P1' scissile bond. The results were shown in **Figure R11D**.

Figure R11. PCSK1 could cleave PRSS35. (A) Western blot analysis of FL-PRSS35 and SF-PRSS35 protein levels when knocking down nine PCSKs using their specific shRNAs. Ponceau staining served as a loading control. FL: full length. SF: short form. (B) Quantitative real-time PCR analysis of nine PCSKs mRNA levels to confirm the knockdown efficiency of their specific shRNAs. (C) PCSK1 protein levels were determined by western blot using the paired human HCC tissues (T) and adjacent non-cancerous liver tissues (N). Ponceau staining and calnexin served as loading controls. (D) Frequency plot of the cleavage sequence in PRSS35 by PCSK1 in a Weblogo format. The size of the symbol indicates the frequency of the individual residue occurrence at individual substrate positions relative to the P1-P1' scissile bond.

Comments 2-6:

6- For the CXCL2 cleavage at the di-basic residue as a trypsin like protein. It is necessary to demonstrate that PRSS35 is acting as a trypsin and the use of selective inhibitors and perform mutagenesis at the catalytic site level.

Response 2-6: Following the reviewer's suggestion, we determined whether PRSS35 could cleave its substrate β -casein in the presence of serine protease inhibitor (Trypsin is a serine protease). The results showed that PRSS35 lost the ability to cleave β -casein in

the presence of serine protease inhibitors, PMSF or protease inhibitor cocktail (**Figure R12A**). These data proved that PRSS35 is an active trypsin like serine protease. Thus, the His-Asp-Ser triad should be the structural basis for the PRSS35 protease activity (Hedstrom 2002). Analysis of PRSS35-Domain1 sequence revealed the potential catalytic sites in PRSS35 based on His-Asp-Ser triad (**Figure R12B**). Importantly, mutation of each serine residue in those sites demonstrated that serine 117 was the catalytic site of PRSS35 (**Figure R12C**). We have included **Figure R12A and C** as **Extended figures 2f and 2g** in the revised manuscript.

Figure R12. Identification of the catalytic site of PRSS35 serine protease activity. (A) *E. coli* purified His-D1 and β -casein protein were incubated with DMSO, serine protease inhibitor 1 (PMSF) or serine protease inhibitor 2 (cocktails) at 37°C overnight, followed by SDS-PAGE and coomassie brilliant blue staining. His-D1 signal was determined by western blot with anti-His antibody. D1: PRSS35 domain1, Serine protease inhibitor 2 (cocktails) included Aprotinin, Bestatin, Leupetin, Pepstatin, PMSF, E-64, and Phosphoramidon. (B) Prediction of possible catalytic site(s) of PRSS35 serine protease activity in its Domain1 region. Colored H, D and S were the possible catalytic relevant amino acids. Usually, serine is the catalytic amino acid. (C) *E. coli* purified wild type His-D1 or mutated His-D1 protein (serine to alanine mutation) as indicated was incubated with β -casein protein at 37°C overnight, followed by SDS-PAGE and coomassie brilliant blue staining. wild type His-D1 and His-D1-mutation signals were determined by western blot with anti-His antibody. Please see also **Extended figures 2f and 2g** in the revised manuscript.

Reviewer #3 (Remarks to the Author):

In this manuscript by Wang et al, the authors employed a proteomic analysis of the secretome of HCC cells and identified a close relationship between the secreted protein PRSS35, NETs and HCC prognosis. Their findings suggest a potential value for PRSS35 as a biomarker for screening in early stage HCC. The main mechanism identified include HCC tumors cell have decreased secreted PRSS35 levels that lead to an increase in the chemokine CXCL2. CXCL2 results in greater neutrophil accumulation and increased NETs formation in the TME which is pro-tumorigenic.

Overall, the authors have performed a considerable body of work, including the analysis of HCC patients, to demonstrate a novel and potentially biomarker role for the secreted protein PRSS35. The manuscript is well written and deeply investigates the mechanisms through characterizing the unique features of PRSS35 suppressing HCC progression by inhibiting CXCL2-mediated neutrophil recruitment and NETs formation.

Response: We appreciate the reviewer for the encouraging comments that have well summarized the significance of our major findings.

Comments 3-1:

1. The main conclusion is that PRSS35 suppresses HCC progression by inhibiting CXCL2-mediated neutrophil recruitment and NETs formation. However, current single-cell technologies show that neutrophils probably have a dynamic spectrum of pro- and anti-tumor (mostly N1 or High-density neutrophils) functions that vary according to their microenvironment (Nat Rev Gastroenterol Hepatol. 2022 Apr;19(4):257-273.). CXCR2+ is a maturation and activation marker of N1 neutrophils which was thought to play an anti-tumor role (Cell Rep. 2015 Feb 3;10(4):562-73.; Front Immunol. 2019 Aug 14;10:1912.; Nat Rev Cancer. 2020 Sep;20(9):485-503.). In this regard, the authors should supplement further analysis of which subpopulation of neutrophil was involved in this protective effect of PRSS35. The anti-tumor role of neutrophils should also be added in the discussion.

Response 3-1: We appreciate the reviewer for the important comments. Following the suggestion, we performed new mouse experiments to analyze the subpopulation of neutrophils involved in the protective effect of PRSS35. Consistent with **Figure 4c** in the original manuscript, mCXCL2 promoted HCC tumor growth in mouse, which was

attenuated by mPRSS35 overexpression (**Figure R13A**). Consistently, enhanced neutrophil recruitment into mouse tumors was observed in mCXCL2 overexpressing group, which was abolished by co-expression with mPRSS35 (**Figure R13B, R13C**). We further analyzed the CXCR2⁺ subpopulation of neutrophils, and found that the number of CXCR2⁺ neutrophils changed along with the total number of neutrophils and the relative CXCR2⁺ neutrophils to the total neutrophils remained the same among different groups (**Figure R13B, R13D**). These data also suggested that the NETs, no matter which kinds of neutrophils it came from, plays important role in the development of HCC. However, as the reviewer mentioned, neutrophils probably have a dynamic spectrum of pro- and anti-tumor functions that vary according to their microenvironment (Geh, Leslie et al. 2022). For example, some studies reported that CXCR2⁺ is a maturation and activation marker of N1 neutrophils which was thought to play anti-tumor roles (Sagiv, Michaeli et al. 2015, Mackey, Coffelt et al. 2019, Jaillon, Ponzetta et al. 2020). However, some recent studies have reported that targeting CXCR2⁺ neutrophils is a promising strategy for the therapy of HCC and PDAC (Nywening, Belt et al. 2018, Kotsiliti 2022). Therefore, the function of subpopulation of neutrophils remains elusive and warrants further study. We have also discussed the anti-tumor role of neutrophils in the discussion section of the revised manuscript.

Figure R13. PRSS35 inhibits HCC tumor development by suppressing neutrophil function. (A) Hepa1-6 cells stably expressing EV, mCXCL2, Flag-mPRSS35, or mCXCL2 plus Flag-mPRSS35 were injected subcutaneously into C57 mice. Photograph of tumors at the end of the experiment (left panel). Tumors were extracted and weighted at the end of the experiment (right panel). (B) Representative FACS analysis of total neutrophil and CXCR2⁺ neutrophil in tumors were shown. (C) Statistical analysis of total neutrophil and CXCR2⁺ neutrophils in tumors were shown.

Comments 3-2:

2. Have the authors overexpressed PRSS35 in human HCC cells lines in addition to murine Hepa1-6 cells? Confirming role of PRSS35 in vivo with a xenograft HCC model may be important in addition to the Hepa1-6-murine HCC model.

Response 3-2: Following the suggestion, we generated human HCC HepG2 cell lines that stably expressed hPRSS35 or an empty vector (EV) control and inoculated these lines subcutaneously into Balb/c-nude mice. The results showed that hPRSS35 overexpression significantly suppressed HepG2 tumor growth in mice (**Figure R14A**). WB analysis of tumor tissues also confirmed that hCXCL2 was depleted in the presence of hPRSS35 (**Figure R14B**). Further examination of the tumor tissues by flow cytometry revealed decreased total neutrophil recruitment into tumors in hPRSS35 overexpressing group, but the proportion of CXCR2⁺ neutrophils remained the same (**Figure R14C, R14D**). Both the MPO-DNA complex data from ELISA analysis and H3Cit protein levels by WB analysis confirmed the decrease of NETs in hPRSS35 overexpressing group (**Figure R14B, R14E**). Taken together, these data documented that PRSS35 suppressed human HCC development in mouse xenograft model. Taken together, in addition to the murine Hepa1-6 cell model and YAP-5SA-induced HCC murine model, here we provide additional data to demonstrate that PRSS35 suppressed human HCC development via regulation of neutrophil recruitment in a xenograft model. We have included part of these data as **Figure 3d** in the revised manuscript

Figure R14. PRSS35 suppressed human HCC development in mouse xenograft model. (A) Equal numbers of HepG2 cells overexpressing EV or hPRSS35 were injected subcutaneously into Balb/c-nude mice. Photograph and weight of tumors at the end of the experiment (35 days after

injection). **(B)** Western blot analysis of hCXCL2, hPRSS5 and H3Cit protein levels in tumors. **(C)** Representative FACS analysis of total neutrophils and CXCR2⁺ neutrophils in tumors were shown. **(D)** Statistical analysis of the total neutrophils and CXCR2⁺ neutrophils in tumors were shown. **(E)** Serum NETs levels were measured by MPO-DNA ELISA kit from mice serum samples. Please also see **Figure 3d** in the revised manuscript.

Comments 3-3:

3. What was the rationale for using YAP-5SA tail vein injection induced HCC model? Have the authors confirmed the recruitment of neutrophils and NET formation in control and PRSS35 KO mice undergoing YAP-5SA HCC model in addition to just measuring number of tumors?

Response 3-3: We appreciate your important questions.

(1) Previous reports have demonstrated that hydrodynamic tail vein injection (Liu, Song et al. 1999) of plasmids is a convenient approach for genetic modification in the mouse liver (Chen and Calvisi 2014). Later, Guo et al found that activation of the Hippo pathway effector Yes-associated protein (YAP) in hepatocytes could recruit M2 macrophage and further initiate HCC development (Guo, Zhao et al. 2017). This group constructed a piggyBac (PB) transposon element (Ding, Wu et al. 2005) expressing human YAP-5SA, an active mutant of YAP, and co-injection of the transposon with a PB transposase plasmid via hydrodynamic tail vein injection led to the accumulation of M2 macrophage and further the initiation of HCC. Therefore, we considered that this model is an efficient and useful model to induce HCC. Indeed, our group have also employed this model in other HCC studies before (Zhang, Sun et al. 2022).

(2) In **Extended Figure 4c-e** and **Figure 4g** in the original manuscript (also see **Figure R15A-D**), we have confirmed the accumulation of neutrophils and NETs in YAP-5SA induced HCC in PRSS35-KO mice compared to that of the WT mice. And the accumulated neutrophils and NETs in YAP-5SA induced HCC in PRSS35-KO mice were depleted by co-injection of shRNAs targeting mCXCL2.

Figure R15. PRSS35 suppressed CXCL2-mediated neutrophil NETS formation in YAP-5SA HCC model. (A-D) The Extended Figure 4c-e and Figure 4g in the original manuscript.

Comments 3-4:

4. The finding that there is less secreted PRSS35 in malignant cells is novel and of great potential for future biomarker. What is the underlying mechanism/regulation of PRSS35 in normal/cancer/cancer cells?

Response 3-4: This is a very good point. Since our data showed that the mRNA levels of PRSS35 was decreased in HCC compared to normal liver tissues, suggesting the transcriptional regulation of PRSS35 in HCC. To uncover the underlying regulation mechanism of PRSS35, we predicted the possible transcriptional factor (s) regulating PRSS35 by JASPAR data base, and found that HNF4A is the most potential candidate. Furthermore, we determined PRSS35 protein and mRNA levels when overexpressing or knocking down HNF4A in HepG2 cells. The results showed that both PRSS35 protein and mRNA levels were down-regulated when we knocked down HNF4A, and up-regulated with HNF4A overexpression (**Figure R16A, B**). In addition, we also predicted the response elements of HNF4A in PRSS35 promoter (**Figure R16C**), and further confirmed these response elements through luciferase assay. The results showed that all the four predicted HNF4A response elements (especially elements 1 and 3) in PRSS35 promoter were responsible for the transcription of PRSS35 (**Figure R16D**). These data are consistent with the previous report that HNF4 had low expression in HCC

(Ning, Ding et al. 2010, Ning, Ding et al. 2014, Cheng, He et al. 2019). Taken together, our data demonstrate that HNF4A is the transcriptional factor regulating PRSS35. We have included these data as **Extended Figures 1i-k** in the revised manuscript.

Figure R16. HNF4A regulates PRSS35 transcription. (A) Western blot analysis of PRSS35 protein when knocking down HNF4A with different shRNAs or overexpressing HNF4A in HepG2 cells. β-actin served as a loading control. FL: full length. SF: short form (B) Quantitative real-time PCR analysis of PRSS35 mRNA levels when knocking down HNF4A with different shRNAs or overexpressing HNF4A in HepG2 cells. (C) A diagram shows the sites and sequences of potential HNF4A responsive elements (HREs) in PRSS35 gene. (D) Luciferase assays were performed to identify HREs in PRSS35 gene. *P < 0.05 compared between the indicated groups. Please see also **Extended Figures 1i-k** in the revised manuscript.

Comments 3-5:

5. The author mentioned that PRSS35 was the most significantly down-regulated protein in the PLC secretome among the 236 secreted proteins. Why did the authors focus on the down-regulated ones rather than up-regulated proteins?

Response 3-5: As shown in **Figure 1a** in the original manuscript, PRSS35 is not only the most significantly down-regulated protein, but also the most significantly changed protein whose fold change outclassed other proteins among both the down- and up-regulated proteins. Furthermore, the gaps in our understanding of PRSS35 and its potential functions have led us to focus on this down-regulated protein.

Comments 3-6:

6. The author mentioned that “they use ELISA to analyze the serum from 149 HCC patients and 73 normal subjects” and the numbers for the information of HCC patients as well as healthy donors in supplementary Table 4 are 149 and 73. However, the number of HCC patients for clinicopathological characteristics in supplementary table2 is 158. Please carefully clarify this mismatch.

Response 3-6: Thank you for the important comments. We apologize for not presenting this information in a clear way in the original manuscript. The **Supplementary Table 2** documented the clinicopathological characteristics of the 158 HCC patients used for the IHC assay (**Extended Figure 1e** in the original manuscript). And the **Supplementary Table 6** documented the basic characteristics of the HCC patients and normal subjects whose sera were used for the ELISA assay (**Figure 1f** in the original manuscript). That is, we used two different batches of clinical patient specimens for the IHC analysis and the ELISA assay. We have specifically stated the two different batches of HCC patient samples in the **Material and Method** section of the revised manuscript.

Reviewer #4 (Remarks to the Author):

This manuscript by Wang et al. reports the discovery of PRSS35 serine protease as a secreted protein that is downregulated in HCC and acts as a tumor suppressor through downregulation of the chemokine CXCL2. Overall, the authors provide convincing evidence that PRSS35 is involved in HCC through in vitro and in vivo experiments. I feel that this manuscript is suitable for publication after the following issues have been addressed:

Response: We appreciate the referee for the positive comments. Meanwhile, we thank the referee for the suggestions and, accordingly, we have provided multiple additional data to strengthen our conclusions in the revised manuscript.

Comments 4-1:

1. The authors used a single cell line representing normal hepatocytes. They should test other normal hepatocyte cell lines as well as primary hepatocytes as well to confirm their findings in a larger set of samples.

Response 4-1: Following the suggestion, we also determined the PRSS35 levels in three batches of primary hepatocytes isolated independently. Our results showed that PRSS35 had higher expression in primary hepatocytes compared to liver cancer cell lines (**Figure R17A**), further confirming that PRSS35 was decreased in liver cancer.

Figure R17 PRSS35 was down-regulated in liver cancer cell lines compared to primary hepatocytes. (A) Western blot analysis of intracellular and extracellular PRSS35 protein levels using N-PRSS35 antibodies in three batches of primary hepatocytes isolated independently, PLC, HepG2, and Hep3B cells.

Comments 4-2:

2. The authors do not provide coverage of the different versions of the PRSS35 protein that they obtained by MS/MS based peptide sequencing. They just show a 'representative' peptide in Figure 1d. The size difference could also potentially be due to glycosylation as

a major post-translational modification of secreted proteins or because of the epitope for antibody-based detection being altered by glycosylation.

Response 4-2: It is an important point. Due to the limitations of LC-MS technology, it is difficult to acquire all the peptides information of different versions of the PRSS35. As the reviewer has pointed out, glycosylation is a common modification of secreted proteins, which could lead to a significant increase of protein molecular weight since the glycosylated groups are usually huge (Eichler 2019). To clarify whether the size difference of PRSS35 was due to the glycosylation modification, we first investigated the sequence of PRSS35 and found that its site 90 asparagine was predicted as a potential site for N-linked glycosylation by UNIPROT database. We further performed immunoprecipitation of PRSS35 using cell lysate and culture media from PLC cells over-expressing PRSS35 with N-PRSS35 or M-PRSS35 antibodies (C-PRSS35 antibody cannot be used for immunoprecipitation), followed by trypsin digestion of the immunoprecipitate and enrichment of the N-linked glycosylation peptides with lectin. After that, we added PNGase into the mixture and collected the digested peptides with C-18 column. The eluted peptides were analyzed by LC-MS. As a result, no PRSS35 peptide was identified by LC-MS, indicating that PRSS35 was not modified by N-linked glycosylation, or its N-linked modification was too weak to detect. Considering our results that PRSS35 size was changed by mutation of PCs cleavage sites (**Figure 2e** in the original and revised manuscript), we could conclude that the size difference in PRSS35 protein was due to the cleavage by PCs.

Comments 4-3:

3. Overall, there is excessive reliance on western blotting to elucidate the nature of the different forms - I suggest that the author use LC-MS/MS analysis to characterize all isoforms that they detected by western blotting. This includes the forms that they see in serum (Figure 1e) as well as in cell line experiments (Figure 2e).

Response 4-3: This is a very good suggestion. Accordingly, we performed immunoprecipitation of PRSS35 using the mixture of cell lysate and culture media from PLC cells over-expressing PRSS35 with the mixed antibodies of anti-N-PRSS35 and anti-M-PRSS35, followed by electrophoresis in SDS-PAGE (**Figure R18A**). The indicated protein bands (1~4) in **Figure R18A** were collected and further analyzed by

LC-MS, respectively. Although it's impossible to acquire all the peptides information of different versions of the PRSS35 due to the limitation of LC-MS technology, we did successfully identify some PRSS35 peptides from these isolated protein bands from gel and found that their sizes were consistent with the in-gel protein bands we observed (Figure R18B).

Figure R18. Identification of the PRSS35 peptides in gels by LC-MS. (A) PRSS35 was immunoprecipitated from the mixture of cell lysate and culture media from PLC cells over-expressing PRSS35 using the mixed antibodies of anti-N-PRSS35 and anti-M-PRSS35, followed by SDS-PAGE separation. (B) The indicated protein bands in A were purified and further analyzed by LC-MS, respectively. The identified peptides from each band by LC-MS were shown (upper panel). The bottom depicts the structure and antibody recognition sequences of N-PRSS35 and M-PRSS35.

Comments 4-4:

4. I have an issue with the database search strategy used by the authors. In the methods (lines 534-535), it seems that they used only a subset of the human protein database. This is wrong for two reasons: i) many proteins that are not secreted can be found in the secretome because of a small number of cells dying; ii) many proteins that are membrane

bound can be found in the secretome; and, iii) many proteins that are secreted as not properly annotated as 'secreted.' The authors should search against the entire human protein database.

Response 4-4: Thank you for pointing this out. Actually, we did use the entire human protein database to search the proteins in the cell culture medium (please see **Extended Figure. 1a** in the original manuscript). However, just as the reviewer pointed out, many proteins that are not secreted may be found in the secretome because of a small number of cells dying or other reasons (e.g. non-classical secreted pathway, enzymolysis of membrane proteins). The concentrations of this kind of proteins are usually very low and the quantification of these proteins by LC-MS would be difficult, which may lead to false positive results. Thus, referring to the strategy used by other groups on the study of the secreted proteins (Deshmukh, Peijs et al. 2019), we decided to exclude these proteins by using a subset of the human protein database documented as “secreted” in Uniprot.

Reference:

Chen, X. and D. F. Calvisi (2014). "Hydrodynamic transfection for generation of novel mouse models for liver cancer research." The American Journal of Pathology **184**(4): 912-923.

Cheng, Z., Z. He, Y. Cai, C. Zhang, G. Fu, H. Li, W. Sun, C. Liu, X. Cui, B. Ning, D. Xiang, T. Zhou, X. Li, W. Xie, H. Wang and J. Ding (2019). "Conversion of hepatoma cells to hepatocyte-like cells by defined hepatocyte nuclear factors." Cell Research **29**(2): 124-135.

Deshmukh, A. S., L. Peijs, J. L. Beaudry, N. Z. Jespersen, C. H. Nielsen, T. Ma, A. D. Brunner, T. J. Larsen, R. Bayarri-Olmos, B. S. Prabhakar, C. Helgstrand, M. C. K. Severinsen, B. Holst, A. Kjaer, M. Tang-Christensen, A. Sanfridson, P. Garred, G. G. Privé, B. K. Pedersen, Z. Gerhart-Hines, S. Nielsen, D. J. Drucker, M. Mann and C. Scheele (2019). "Proteomics-Based Comparative Mapping of the Secretomes of Human Brown and White Adipocytes Reveals EPDR1 as a Novel Adipokine." Cell metabolism **30**(5).

Ding, S., X. Wu, G. Li, M. Han, Y. Zhuang and T. Xu (2005). "Efficient transposition of the piggyBac (PB) transposon in mammalian cells and mice." Cell **122**(3): 473-483.

Eichler, J. (2019). "Protein glycosylation." Current Biology : CB **29**(7): R229-R231.

- Geh, D., J. Leslie, R. Rumney, H. L. Reeves, T. G. Bird and D. A. Mann (2022). "Neutrophils as potential therapeutic targets in hepatocellular carcinoma." Nature Reviews. Gastroenterology & Hepatology **19**(4): 257-273.
- Guo, X., Y. Zhao, H. Yan, Y. Yang, S. Shen, X. Dai, X. Ji, F. Ji, X.-G. Gong, L. Li, X. Bai, X.-H. Feng, T. Liang, J. Ji, L. Chen, H. Wang and B. Zhao (2017). "Single tumor-initiating cells evade immune clearance by recruiting type II macrophages." Genes & Development **31**(3): 247-259.
- Hedstrom, L. (2002). "Serine protease mechanism and specificity." Chemical Reviews **102**(12): 4501-4524.
- Jaillon, S., A. Ponzetta, D. Di Mitri, A. Santoni, R. Bonecchi and A. Mantovani (2020). "Neutrophil diversity and plasticity in tumour progression and therapy." Nature reviews. Cancer **20**(9): 485-503.
- Kotsiliti, E. (2022). "CXCR2 inhibition in NASH-HCC." Nature Reviews. Gastroenterology & Hepatology **19**(7): 415.
- Leslie, J., J. B. G. Mackey, T. Jamieson, E. Ramon-Gil, T. M. Drake, F. Fercoq, W. Clark, K. Gilroy, A. Hedley, C. Nixon, S. Luli, M. Laszczewska, R. Pinyol, R. Esteban-Fabro, C. E. Willoughby, P. K. Haber, C. Andreu-Oller, M. Rahbari, C. Fan, D. Pfister, S. Raman, N. Wilson, M. Müller, A. Collins, D. Geh, A. Fuller, D. McDonald, G. Hulme, A. Filby, X. Cortes-Lavaud, N.-E. Mohamed, C. A. Ford, X. L. Raffo Iraolagoitia, A. J. McFarlane, M. V. McCain, R. A. Ridgway, E. W. Roberts, S. T. Barry, G. J. Graham, M. Heikenwälder, H. L. Reeves, J. M. Llovet, L. M. Carlin, T. G. Bird, O. J. Sansom and D. A. Mann (2022). "CXCR2 inhibition enables NASH-HCC immunotherapy." Gut.
- Liu, F., Y. Song and D. Liu (1999). "Hydrodynamics-based transfection in animals by systemic administration of plasmid DNA." Gene Therapy **6**(7): 1258-1266.
- Mackey, J. B. G., S. B. Coffelt and L. M. Carlin (2019). "Neutrophil Maturity in Cancer." Frontiers In Immunology **10**: 1912.
- Ning, B.-F., J. Ding, J. Liu, C. Yin, W.-P. Xu, W.-M. Cong, Q. Zhang, F. Chen, T. Han, X. Deng, P.-Q. Wang, C.-F. Jiang, J.-P. Zhang, X. Zhang, H.-Y. Wang and W.-F. Xie (2014). "Hepatocyte nuclear factor 4 α -nuclear factor- κ B feedback circuit modulates liver cancer progression." Hepatology (Baltimore, Md.) **60**(5): 1607-1619.
- Ning, B.-F., J. Ding, C. Yin, W. Zhong, K. Wu, X. Zeng, W. Yang, Y.-X. Chen, J.-P. Zhang, X. Zhang, H.-Y. Wang and W.-F. Xie (2010). "Hepatocyte nuclear factor 4 alpha suppresses the development of hepatocellular carcinoma." Cancer Research **70**(19): 7640-7651.
- Nywening, T. M., B. A. Belt, D. R. Cullinan, R. Z. Panni, B. J. Han, D. E. Sanford, R. C.

Jacobs, J. Ye, A. A. Patel, W. E. Gillanders, R. C. Fields, D. G. DeNardo, W. G. Hawkins, P. Goedegebuure and D. C. Linehan (2018). "Targeting both tumour-associated CXCR2 neutrophils and CCR2 macrophages disrupts myeloid recruitment and improves chemotherapeutic responses in pancreatic ductal adenocarcinoma." Gut **67**(6): 1112-1123.

Remacle, A. G., S. A. Shiryaev, E.-S. Oh, P. Cieplak, A. Srinivasan, G. Wei, R. C. Liddington, B. I. Ratnikov, A. Parent, R. Desjardins, R. Day, J. W. Smith, M. Lebl and A. Y. Strongin (2008). "Substrate cleavage analysis of furin and related proprotein convertases. A comparative study." The Journal of Biological Chemistry **283**(30): 20897-20906.

Sagiv, J. Y., J. Michaeli, S. Assi, I. Mishalian, H. Kisos, L. Levy, P. Damti, D. Lumbroso, L. Polyansky, R. V. Sionov, A. Ariel, A.-H. Hovav, E. Henke, Z. G. Fridlender and Z. Granot (2015). "Phenotypic diversity and plasticity in circulating neutrophil subpopulations in cancer." Cell Reports **10**(4): 562-573.

Tian, S., W. Huajun and J. Wu (2012). "Computational prediction of furin cleavage sites by a hybrid method and understanding mechanism underlying diseases." Scientific Reports **2**: 261.

Wang, H., H. Zhang, Y. Wang, Z. J. Brown, Y. Xia, Z. Huang, C. Shen, Z. Hu, J. Beane, E. A. Ansa-Addo, H. Huang, D. Tian and A. Tsung (2021). "Regulatory T-cell and neutrophil extracellular trap interaction contributes to carcinogenesis in non-alcoholic steatohepatitis." Journal of Hepatology **75**(6): 1271-1283.

Yang, L.-Y., Q. Luo, L. Lu, W.-W. Zhu, H.-T. Sun, R. Wei, Z.-F. Lin, X.-Y. Wang, C.-Q. Wang, M. Lu, H.-L. Jia, J.-H. Chen, J.-B. Zhang and L.-X. Qin (2020). "Increased neutrophil extracellular traps promote metastasis potential of hepatocellular carcinoma via provoking tumorous inflammatory response." Journal of hematology & oncology **13**(1): 3.

Zhang, T., L. Sun, Y. Hao, C. Suo, S. Shen, H. Wei, W. Ma, P. Zhang, T. Wang, X. Gu, S.-T. Li, Z. Chen, R. Yan, Y. Zhang, Y. Cai, R. Zhou, W. Jia, F. Huang, P. Gao and H. Zhang (2022). "ENO1 suppresses cancer cell ferroptosis by degrading the mRNA of iron regulatory protein 1." Nature Cancer **3**(1): 75-89.

REVIEWERS' COMMENTS

Reviewer #1 (Remarks to the Author):

The authors have answered successfully all the points that I originally raised. I do not understand why they left out two important results.

1.- In NASH, PRSS35 also goes down.

2- PRSS35 can degrade some CXCR chemokines and other are not degraded by this protease.

I leave to the editorial team and the authors to decide if these results are to be included, by I particularly think they are very relevant results.

Reviewer #2 (Remarks to the Author):

The authors have perfectly satisfy to the comments.

Thus, I accept the revised manuscript.

With my best regards

Pr.M.Salzet

Reviewer #3 (Remarks to the Author):

The authors have adequately responded to my previous comments and have performed a significant number of new experiments to satisfy my previous concerns. I have no further questions.

Allan Tsung

REVIEWERS' COMMENTS

Reviewer #1 (Remarks to the Author):

Comment 1-1:

The authors have answered successfully all the points that I originally raised. I do not understand why they left out two important results.

1.- In NASH, PRSS35 also goes down.

2- PRSS35 can degrade some CXCR chemokines and other are not degraded by this protease.

I leave to the editorial team and the authors to decide if these results are to be included, by I particularly think they are very relevant results.

Response: We appreciate the referee's positive comments. We agree with our reviewer that these data are important. However, given the length of manuscript and focus of the current study, we decided not to include these results.

Reviewer #2 (Remarks to the Author):

Comment 2-1:

The authors have perfectly satisfy to the comments.

Thus, I accept the revised manuscript.

With my best regards

Pr.M.Salzet

Response: We appreciate the referee's positive comments.

Reviewer #3 (Remarks to the Author):

Comment 3-1:

The authors have adequately responded to my previous comments and have performed a significant number of new experiments to satisfy my previous concerns. I have no further questions.

Allan Tsung

Response: We appreciate the referee's positive comments.